

# Publishing statistical models:
# Getting the most out of particle physics experiments

Kyle Cranmer[1†⋆], Sabine Kraml[2‡⋆], Harrison B. Prosper [3∘⋆], Philip Bechtle[4],
Florian U. Bernlochner[4], Itay M. Bloch[5], Enzo Canonero[6], Marcin Chrzaszcz[7],
Andrea Coccaro[8], Jan Conrad[9], Glen Cowan[10], Matthew Feickert[11],
Nahuel F. Iachellini[12,13], Andrew Fowlie[14], Lukas Heinrich[15], Alexander Held[1],
Thomas Kuhr[13,16], Anders Kvellestad[17], Maeve Madigan[18], Farvah Mahmoudi[15,19],
Knut D. Morå[20], Mark S. Neubauer[11], Maurizio Pierini[15], Juan Rojo[8], Sezen Sekmen[22],
Luca Silvestrini[23], Veronica Sanz[24,25], Giordon Stark[26], Riccardo Torre[8],
Robert Thorne[27], Wolfgang Waltenberger[28], Nicholas Wardle[29] and Jonas Wittbrodt[30]

## Abstract

The statistical models used to derive the results of experimental analyses are of incredible scientific value and are essential information for analysis preservation and reuse. In this paper, we make the scientific case for systematically publishing the full statistical models and discuss the technical developments that make this practical. By means of a variety of physics cases — including parton distribution functions, Higgs boson measurements, effective field theory interpretations, direct searches for new physics, heavy flavor physics, direct dark matter detection, world averages, and beyond the Standard Model global fits — we illustrate how detailed information on the statistical modelling can enhance the short- and long-term impact of experimental results.



**1** New York University, USA
**2** LPSC Grenoble, France
**3** Florida State University, USA
**4** University of Bonn, Germany
**5** School of Physics and Astronomy, Tel-Aviv University, Israel
**6** University of Genova, Italy
**7** Institute of Nuclear Physics, Polish Academy of Sciences, Krakow, Poland
**8** INFN, Sezione di Genova, Italy
**9** Oskar Klein Centre, Stockholm University, Sweden
**10** Royal Holloway, University of London, UK
**11** University of Illinois at Urbana-Champaign, USA
**12** Max Planck Institute for Physics, Munich, Germany
**13** Exzellenzcluster ORIGINS, Garching, Germany
**14** Nanjing Normal University, Nanjing, PRC
**15** CERN, Switzerland
**16** Ludwig-Maximilians-Universität München, Germany
**17** University of Oslo, Norway
**18** DAMTP, University of Cambridge, UK

**19** Lyon University, France
**20** Columbia University 10027, USA
**21** VU Amsterdam and Nikhef, The Netherlands
**22** Kyungpook National University, Daegu, Korea
**23** INFN, Sezione di Roma, Italy
**24** University of Sussex, UK
**25** IFIC, Universidad de Valencia-CSIC, Spain
**26** SCIPP, UC Santa Cruz, CA, USA
**27** University College London, UK
**28** HEPHY and University of Vienna, Austria
**29** Imperial College London, UK
**30** Lund University, Sweden

⋆ Editors
† kyle.cranmer@nyu.edu, ‡ sabine.kraml@lpsc.in2p3.fr, ○ harry@hep.fsu.edu,
(mailing list: open-likelihoods@cern.ch)

# Contents

# 1  Introduction

In 2000, Fred James and Louis Lyons (with considerable help from Yves Perrin) convened an unusual meeting at CERN [1] between physicists and a few statisticians to discuss the somewhat dry topic of confidence limits. There was a great deal of discussion at this workshop about the nature of probability, its relationship to experimental physics, and the fact that every measurement, which unavoidably includes random effects, is based on a statistical model. At the end of the CERN workshop, the first of a series that came to be known as *PhyStat* [2], Bob Cousins summarized the points on which there was broad agreement. One point of agreement was that particle physicists should publish likelihood functions, given their fundamental importance in extracting quantitative results from experimental data. This paper makes the scientific case for publishing not only likelihood functions, but full statistical models — and for making this a standard practice.

The preservation of the experimental results in a readily usable form is now widely accepted as important for the continued progress of particle physics. Indeed, all collaborations at the Large Hadron Collider (LHC) now either require, or strongly encourage, experimental physicists to make their results available in databases such as `HEPData` [3–5]. Many particle theory groups are using this information for wider and/or more general interpretation of the experimental results, and have built public software frameworks for this purpose; see Ref. [6] for an overview of approaches and public tools. These important initiatives build on a bedrock principle of publicly-funded science: the open exchange of scientific results as a way to enable scientific progress and empower those who are sufficiently motivated to contribute to the scientific enterprise.

It is in this spirit that a discussion about making the results of particle physics experiments broadly available was initiated at the *2011 Les Houches Workshop*, resulting in a set of clear-cut recommendations [7]. The discussion has been continued within the *Reinterpretation Forum* [8], with the current status and updated recommendations presented in Ref. [6]. This paper takes these decade-long efforts to what we argue is the logical conclusion: if we wish to maximize the scientific impact of particle physics experiments, decades into the future, we should make the publication of full statistical models, together with the data to convert them into likelihood functions, standard practice. A statistical model provides the complete mathematical description of an experimental analysis and is, therefore, the appropriate starting point for any detailed interpretation of the experimental results. The goal of this paper is to explain through selected physics cases why it would be of great scientific benefit to make full statistical models and the associated data publicly available.

The paper is organized as follows. In Section 2, we begin by clarifying the statistical terminology and concepts used in this paper. The section also includes a discussion of what statistical information and products should be published, and serves as a high-level introduction to Section 3, which discusses technical considerations pertaining to the publication of statistical models as well as descriptions of existing statistical infrastructure and tools. This sets the stage for the main task of this paper, which is to give physics cases that highlight the potential for substantial scientific progress if full statistical models were publicly available and to indicate where progress is currently impeded because they are not yet routinely published. These physics cases, which are presented in Section 4, cover parton distribution functions, Higgs boson measurements, effective field theory (EFT) interpretations, direct searches for beyond the Standard Model (BSM) particles, heavy flavor physics, direct dark matter detection, world averages, and BSM global fits. While certainly not exhaustive, this set of examples is illustrative of the potential wide-ranging scientific impact of publicly available high-fidelity statistical models that encapsulate in detail the mathematical structure of experimental analyses. Finally, Section 5 highlights a few outstanding issues that merit further study, and Section 6 summarizes the paper.

## 2 From statistical models to likelihoods

### 2.1 Basic concepts

The term likelihood and other statistical concepts are sometimes used in a colloquial way by physicists, which can cause confusion. Therefore, we begin with a review of the key ideas pertaining to statistical models, likelihoods and related concepts [9–11] so that it is clear what we mean when we refer to these concepts in this paper.

Consider a collection of values $x$ as the outcome of a measurement, which we treat as a random variable. Statisticians distinguish between a random variable, typically denoted by an upper-case letter, for example, $X$, and a given realization of the random variable, written with the corresponding lower-case letter. For simplicity, however, we do not make that distinction. Here $x$ could represent single or multiple values, and its components can be discrete or continuous. It could be the result of an individual collision event, a set of events, or a collection of multiple datasets.

The statistical nature of the data is quantified by ascribing to it a probability, as specified by a statistical model. Often members of a family of models can be labeled by a single or multidimensional parameter $\omega$, which can be continuous or discrete. We write the probability of the data $x$ under the assumption of $\omega$ as $p(x|\omega)$, with the understanding that this represents a probability density function (pdf) if $x$ is continuous and a probability mass function (pmf) if the data are discrete.[1]

Statistical methods fall into two broad categories depending on how the analyst chooses to interpret probability. In *frequentist* statistics, probability is associated only with the data, $x$, and is interpreted as a limiting frequency for repeated outcomes of the observation. In *Bayesian* statistics, probability is also assigned to the hypothesized model, or to the parameters that label different models, and it represents a degree of belief that the model is true.

When data are entered into a statistical model the resulting function of the parameters is called the likelihood of the hypothesis. Sometimes in frequentist statistics the data $x$ are suppressed in the notation. For example, one often writes the likelihood function as $L(\omega) = p(x|\omega)$. For some frequentist methods, in particular for large data samples, it may be

---

[1]To lighten the prose, we shall use the term probability or probability model as shorthand for pdf or pmf and rely upon the context to make clear which we mean.

sufficient to know only $L(\omega)$ evaluated with the observed $x$. In general, however, one requires the probabilities not only for the actual data but also for hypothetical data that were not observed. Therefore, it is insufficient in many cases to report only the likelihood's dependence on the parameters, and we must give the statistical model $p(x|\omega)$ with the full dependence on both $x$ and $\omega$.

In Bayesian statistics one uses the likelihood $p(x|\omega)$ (the conditional probability for $x$ given $\omega$) with Bayes' theorem to find the probability for the model parameter $\omega$ given the observed data $x$,

$$p(\omega|x) = \frac{p(x|\omega)\pi(\omega)}{\int p(x|\omega)\pi(\omega)\,d\omega} \, . \tag{1}$$

Here the prior probability $\pi(\omega)$ represents one's degree of belief about the parameter before consideration of the data $x$. In Bayesian statistics, all of one's knowledge about the parameter after inclusion of the data is contained in the posterior density $p(\omega|x)$. In contrast to frequentist results, the Bayesian posterior probability only depends on the observed data, not on hypothetical data that were not seen (Bayesian inference is said to obey the likelihood principle [12]).

Out of the full set of parameters represented by $\omega$ one usually distinguishes between those that are of (current) interest, $\mu$, and all the rest, called nuisance parameters $\theta$. The parameters of interest could be, e.g., the rate of a signal process or mass of a particle, while the nuisance parameters may include calibration constants, background model parameters, etc.

Nuisance parameters are generally included in a model to take into account systematic uncertainties. Suppose that $x$ are the *primary measurements* and have probability (density) $p(x|\mu,\theta)$. In order to constrain the nuisance parameters $\theta$ we have a set of independent *auxiliary data* $y$, with probability $p(y|\theta)$. The joint probability of the data $x$ and $y$ is, therefore, given by

$$p(x,y|\mu,\theta) = p(x|\mu,\theta)p(y|\theta) \, . \tag{2}$$

The values $y$ are often estimates of corresponding nuisance parameters, and their probability may be, e.g., a Gaussian with a specified standard deviation. In many cases $y$ are measurements in a control region, although some components of $y$ may not be actual measurements but perhaps estimates based on theoretical predictions implemented in a Monte Carlo (MC) simulation. In the frequentist framework these data enter on the same footing as other data and are treated as such. Equation (2) gives the *full statistical model*, understood to include both the primary measurements $x$ as well as the auxiliary data $y$.

In the Bayesian approach, one usually incorporates the information from any auxiliary data into the prior for the nuisance parameters. First let us suppose that the parameters of interest $\mu$ and nuisance parameters $\theta$ are independent, so that the joint prior factorizes: $\pi(\mu,\theta) = \pi(\mu)\pi(\theta)$. The prior for the nuisance parameters *before* inclusion of the primary measurements but *after* inclusion of the auxiliary data is

$$\pi(\theta) = p(\theta|y) \propto p(y|\theta)\pi_0(\theta) \, . \tag{3}$$

Here $\pi_0(\theta)$ (the "Ur-prior") is the prior for $\theta$ before inclusion even of the auxiliary data. This prior reflects whatever information was available at that point and is often taken to be very broad or even constant, though more sophisticated forms are possible (see, for example, Ref. [13]). Regardless of how $\pi_0(\theta)$ is chosen, the information from the auxiliary data is incorporated into $\pi(\theta)$ and from this point $y$ no longer appears explicitly; that is, $y$ is not treated as part of the data together with the primary measurements $x$. Therefore, to report the full statistical model for use in a Bayesian analysis, one gives the likelihood $p(x|\mu,\theta)$ and the prior $\pi(\theta)$, which encodes the information from the auxiliary data.

If the analyst reports $p(x,y|\mu,\theta)$, then this can still be used in a Bayesian analysis simply by applying the Ur-prior $\pi_0(\theta)$ for the nuisance parameters. If, on the other hand, one reports

the likelihood for the primary measurements $p(x|\mu,\theta)$ and a prior $\pi(\theta)$, then for further use in a frequentist analysis one often averages the statistical model $p(x|\mu,\theta)$ with respect to the prior,

$$p(x|\mu) = \int p(x|\mu,\theta)\,\pi(\theta)\,d\theta\;. \tag{4}$$

In this paper we treat the primary and auxiliary data as part of a single set of data and, therefore, write the statistical model, $p(x,y|\mu,\theta) = p(x|\mu,\theta)p(y|\theta)$, with the full dependence on both the data (primary and auxiliary) and parameters (of interest and nuisance).

A quite general form of Eq. (2), used in particle physics, can be constructed in the following way. Suppose that a set of $N_c$ independent channels for events is defined. The channels could be, for example, distinct physics channels, bins of a (possibly multidimensional) histogram, or signal regions in a search for new physics. A certain number of events $n_i$ is found in channel $i$, and for every event $j$ in that channel one measures a vector of values $x_{ij}$. Suppose the $n_i$ counts can be modeled as independent and Poisson distributed with mean counts $\nu_i(\mu,\theta)$ (typically, the product of integrated luminosity, cross section, branching ratio, efficiency, and acceptance). The mean counts $\nu_i(\mu,\theta)$ are generally a sum

$$\nu_i(\mu,\theta) \;\;=\;\; \sum_k \nu_{ik}(\mu,\theta) \tag{5}$$

of contributions $\nu_{ik}(\mu,\theta)$ from various physics processes (indexed by $k$) that do not interfere quantum mechanically. Each process $k$ is associated with a probability $p_{ik}(x|\mu,\theta)$ to produce the vector of outcomes $x$ for channel $i$. The probability $p_i(x_{ij}|\mu,\theta)$ to measure $x_{ij}$ in channel $i$ event $j$ is hence the weighted sum

$$p_i(x_{ij}|\mu,\theta) \;\;=\;\; \sum_k \frac{\nu_{ik}(\mu,\theta)}{\nu_i(\mu,\theta)} p_{ik}(x_{ij}|\mu,\theta)\,, \tag{6}$$

which statisticians call a mixture model and which corresponds to the familiar "stacked histogram". Given auxiliary data $y$ with pdf $p(y|\theta)$, whose form we leave open, the full statistical model can be written as

$$p(n,x,y|\mu,\theta) = \prod_{i=1}^{N_c}\left[\text{Pois}(n_i\mid\nu_i(\mu,\theta))\prod_{j=1}^{n_i}p_i(x_{ij}|\mu,\theta)\right]p(y|\theta)\to L(\mu,\theta)\,, \tag{7}$$

where $n$, $x$, and $y$ denote all *observable* data and $\to$ signifies that the statistical model becomes the likelihood $L(\mu,\theta)$ when *observed* data are entered into it. In practice, $p(y|\theta)$ often factorizes into independent terms for independent sources of systematic uncertainty. The statistical model in Eq. (7) is quite general in that it covers analyses that use binned data, unbinned data, or a combination of both. Furthermore, it covers both closed-world as well as open-world models as defined in Fig. 1 and Sec. 2.2.

Access to the individual components $\nu_{ik}$ and $p_{ik}$ is not typically needed for the purpose of likelihood combinations; however, it is needed for some reinterpretations of which three types can be distinguished. The first type entails reparametrizing the likelihood, with $\mu \to \mu'(\mu)$, without altering the efficiencies and acceptances that might modify the distributions. This type of *parametric reinterpretation* includes the case where the signal cross section and branching ratios for several components can be related to some new model parameters $\mu'$. The second type requires changing the distributions $p_{ik}(x_{ij}|\mu,\theta)$ because a different physical process with a different phase space distribution is being considered, which might have different efficiencies and acceptances. This type of *kinematic reinterpretation* is relevant, for instance, for reinterpreting direct searches that target new models (i.e., "recasting") and is sometimes necessary

for EFT reinterpretations depending on the details of the original model and target EFT.[2] The third type repeats previous work using more precise theoretical calculations or improved experimental calibrations, or both. For such *updates of (re)interpretations* both the distributions as well as the way a model might be parametrized may change.

In constructing Eq. (7), one builds the term in brackets for each channel taking care to identify shared parameters and to avoid double counting constraint terms defined by the auxiliary data $y$. Therefore, to construct the full statistical model one must have access to, and be able to identify, the individual terms.

Nuisance parameters are treated in different ways in the frequentist and Bayesian approaches. In a frequentist analysis, given a likelihood $L(\mu, \theta)$ that depends on parameters of interest $\mu$ and nuisance parameters $\theta$, one may eliminate the nuisance parameters by constructing the *profile likelihood*,

$$L_{\mathrm{p}}(\mu) = L(\mu, \hat{\hat{\theta}}(\mu)) \,. \tag{8}$$

Here $\hat{\hat{\theta}}(\mu)$ are the profiled nuisance parameters, i.e., the values that maximize the likelihood for given values of the parameters of interest. For analyses based on a sufficiently large data sample it is often possible to eliminate nuisance parameters in this way, so that the final inference about the parameters of interest holds for any values of the nuisance parameters (see, for example, Ref. [16]).

In a Bayesian analysis, in contrast, nuisance parameters are eliminated by *marginalizing* the posterior density to find the probability for the parameters of interest alone,

$$p(\mu|x) = \int p(\mu, \theta|x) \, d\theta \,. \tag{9}$$

This is often a computationally challenging step, which may require, for example, the use of Markov Chain Monte Carlo (MCMC) integration.

In some analyses the parameters of interest represent the expected numbers of entries in bins of a differential distribution. There are two basic approaches to this problem: "unfolding" and "folding". The two approaches lead to different requirements for what must be reported for further analysis.

Often when measuring a distribution one defines parameters $\vec{\mu} = (\mu_1, \ldots, \mu_M)$ to represent the expected number of entries in a bin assuming perfect resolution (so-called particle-level or truth-level parameters). In practice, the real detector has limited resolution and so an event with a true value of a variable in a certain bin might be measured (reconstructed) in a different one, so that $\vec{\nu} = (\nu_1, \ldots \nu_N)$ represents the expected numbers of events at reconstructed or "detector" level. These are related by

$$\vec{\nu} = R\vec{\mu} \,, \tag{10}$$

where $R$ is an $N \times M$ *response matrix* defined such that $R_{ij}$ represents the probability to measure $\nu$ in bin $i$ given that its true value $\mu$ was in bin $j$ (here we neglect background processes).[3]

Estimating the truth-level parameters $\vec{\mu}$ or *unfolding* of the distribution results in correlated estimators. The estimators are often treated as a Gaussian distributed vector characterized by an $M \times M$ covariance matrix $U_{ij} = \mathrm{cov}[\hat{\mu}_i, \hat{\mu}_j]$. In addition, the estimators are often constructed to include a small bias in exchange for a reduction in statistical variance (regularized unfolding). By contrast, in *folding* one reports estimators for the expected numbers of events

---

[2]We note that through "morphing" one can represent the distributions for EFTs as a mixture including the effect of interference [14, 15]. This allows one to convert what would be a kinematic reinterpretation into a parametric reinterpretation, which is more convenient.

[3]Often one assumes that the response matrix $R$ is independent of the true distribution but this is not exact; see, e.g., Ref. [9], Chapter 11.

at the detector level $\vec{v}$. In the simplest case one has $\hat{v}_i = n_i$, where $n_i$ is the observed number of entries in the $i$th bin. Then to compare this result to the prediction of a certain model that predicts a particle-level distribution $\vec{\mu}$, one needs to "fold" the model prediction with the response matrix, i.e., one compares the $\vec{n} = (n_1, \ldots, n_N)$ to the $\vec{v}$ from Eq. (10). Often this likelihood will treat the $n_i$ as independent and Poisson distributed.

An advantage of unfolding is that the estimated parameters represent directly the distribution in question. They can be compared between experiments and to model predictions using, e.g., a multivariate Gaussian likelihood for which the covariance matrix of the estimators $U$ is essential. A disadvantage is that the unfolding may require regularization, which necessarily introduces some bias. In folding, one simply reports the observed numbers of events in the bins, and thus no regularization bias enters. But to compare these results with a model prediction, one needs the response matrix $R$. A further disadvantage of unfolding is that the covariance matrix will usually include contributions from systematic uncertainties. If unfolded estimates are produced for several distributions, then the information on correlations between bins of different distributions is difficult to reconstruct, and thus it becomes difficult to combine the distributions in a global fit. In folding, however, one can retain information about common systematic effects provided that the response matrices for the different distributions are supplied as a function of the corresponding common nuisance parameters.

This concludes our discussion of basic concepts. For convenience and clarity, we summarize in Fig. 1 the definitions of a few key terms used throughout this paper.

## 2.2 High-level considerations on what to publish

In the preceding section we discussed the underlying statistical concepts needed to understand what information is necessary to enable various types of usage. Now we transition to high-level considerations on what the experimental collaborations might publish. A few questions drive the bulk of the discussion.

- **Functionality:** Does one publish the likelihood function $L(\mu, \theta)$, a profile likelihood $L_\mathrm{p}(\mu)$, or the full statistical model $p(x, y \mid \mu, \theta)$ together with the observed data $x$ and $y$? Also, to what extent should, or can, the parameter vector $\mu$ be independent of the physics model?

- **Preservation:** Is the statistical model closed-world or open-world?

- **Data Representation:** Does the model describe binned or unbinned data?

When considering down-stream functionality, the full statistical model together with the corresponding data are the gold standard as they enable combinations, reinterpretations, and the generation of synthetic or pseudo data ("toy Monte Carlo") that are typically needed for frequentist procedures (see e.g. Section 4.2.2), or for validating statistical procedures. As mentioned in the previous section, it is often necessary to have access to individual terms in the statistical model, e.g., to avoid double counting constraint terms in combinations or for reinterpretations that require modifications to individual signal or background components.

In contrast, publishing the profile likelihood $L_\mathrm{p}(\mu)$ or the full likelihood $L(\mu, \theta)$ can be convenient, but precludes the generation of pseudo data, and renders combinations problematic because of issues with double counting constraint terms and/or inconsistent profiling of the nuisance parameters. In any case, care needs to be taken to ensure a sensible parametrization, as discussed also in some of the physics cases in Section 4. In particular, likelihood parametrization in terms of physical (pseudo) observables like masses, cross-sections, widths, branching fractions, etc., is often more useful than a parametrization in terms of theory-model

> ## Glossary of terms
>
> - **Statistical model**: This is a synonym for the probability model $p(x, y|\mu, \theta)$ as in Eq. (7) that includes dependence on the data $x$ and $y$, the parameters of interest $\mu$ and nuisance parameters $\theta$, access to the individual terms and the ability to generate pseudo- (or synthetic-) data (i.e., "toy Monte Carlo").
>
> - **Likelihood**: The value of the statistical model for a given *fixed* dataset as a function of the parameters, e.g., $L(\mu, \theta)$ in Eq. (7).
>
> - **Constraint term**: A term in the full statistical model that relates auxiliary data $y$ to a particular nuisance parameter $\theta$.
>
> - **Observed data** the $n$, $x$, and $y$ of Eq. (7) needed to construct the likelihood.
>
> - **Open-world**: An approach to statistical modelling that allows users to define and implement custom components in the statistical model.
>
> - **Closed-world**: An approach to statistical modelling that requires users to work with a finite set of modelling components.
>
> - **Declarative specification**: An unambiguous specification (e.g., of a statistical model) that is independent of implementation. Often there exists a reference implementation of a specification, but in the declarative approach there may be multiple implementations that are conceptually and mathematically equivalent.
>
> - **Serialization**: The process of writing a data structure (e.g., a statistical model) in memory to a file in a way that can be read back into memory. Loading the serialized object typically requires access to compatible software libraries present at the time of serialization.

Figure 1: Definitions of a few key terms used in this paper.

(Lagrangian) parameters like couplings or $\tan \beta$. It is, therefore, recommended that likelihoods be published in addition to, not instead of, the full statistical model.

When considering technical aspects of preservation of a full statistical model, such as that in Eq. (7), the dominant consideration is whether the statistical model is unrestricted (open-world) in its implementation or not (closed-world). In open-world approaches users can extend the modelling components, but this has down-stream technical ramifications. In a closed-world approach, there is typically a modelling language with a finite number of component building blocks, which simplifies the development of a declarative specification for a statistical model and the associated data. We discuss these matters in detail in Sections 3.1 and 3.2.

Next, we come to the analysis design. One of the first questions one might ask about an analysis is whether it uses binned or unbinned data. As noted earlier, Eq. (7) is quite general. One can relate binned and unbinned data by thinking of the probability model $p_i(x_{ij}|\mu, \theta)$ as piecewise constant, and thus one need only know the number of events observed in any given bin and the corresponding probability (which can then be recast into the form of a product of Poisson terms). From this perspective, there is no fundamental difference between binned and unbinned models, but in practice it easier to define a closed-world declarative specification for binned analyses than for unbinned ones.

Based on these considerations, which we expand upon below, we recommend that when

possible three data products be published:

1. the observed data (*e.g.* the $n$, $x$, and $y$ of Eq. (7)) ,

2. the statistical model,

3. and if applicable a set of additional model fragments to enable kinematic and other types of reinterpretation.

The statistical model should include as distinct, identifiable components, those pertaining to the primary measurements $p_{ik}(x_i|\mu, \theta)$ as well as the terms $p(y|\theta)$ pertaining to the auxiliary data as described in Eqs. (5)–(7). Using the published data alone, it should thus be possible to reproduce inferences of the original analysis, such as maximum-likelihood estimates of parameters, or their posterior densities.

Under this broad recommendation, we note that there are two actions that can be taken immediately.

1. The first immediate action is to publish the `RooWorkspace` [17] of the models used for current (and past) publications. As discussed in Section 3.1 there are drawbacks to the `RooWorkspace`; however, we recognize that this is the method of specification and serialization for the vast majority of particle physics statistical models. Their publication would have great value for analysis and data preservation and would signal future intentions.

2. The second immediate action is relevant to the statistical models based on the `HistFactory` specification. Here we recommend that these models be converted into the JSON format introduced by `pyhf` [18,19] and published to `HEPData` [4] as was done by ATLAS in Refs. [20–27].

In this early phase, it is natural that the focus be on closed-world binned analyses as they lend themselves directly to scientific use cases that may quickly demonstrate the value of these data products, establish new practices, and foster activities that improve the related infrastructure. In the longer term, once the publication of closed-world binned statistical models becomes routine, the experience gained from the public availability of these data products may inform the development of a larger set of statistical models.

## 3 Technical considerations

The focus of this section is the technical aspects of publishing full statistical models and likelihoods. We discuss the tools and general considerations around serialization and down-stream use of statistical models, but not the many tools that exist for the construction and evaluation of statistical models. The discussion is separated into the open-world and closed-world approaches and approaches that use machine learning to represent the likelihood or a closely related quantity. However, before getting into those details, let us set the stage and frame the discussion.

While many of our considerations are independent of the specific technology, it is important to recognize that by far the most common tool in particle physics for statistical modelling is `RooFit` [28], which is distributed with `ROOT`. `RooFit` provides the core interfaces, modelling language (needed to combine various terms as in Eqs. (6) and (7)), as well as a large library of modelling components (needed to represent the individual distributions $p_{ik}(x|\mu, \sigma)$). The library of modelling components that `RooFit` provides is finite, but large enough that it stretches the intent of the term closed-world. Moreover, users can extend the library by

inheriting from the `RooAbsPdf` abstract interface, which enables an open-world approach to modelling if desired. In addition to `RooFit`, the software package `RooStats` builds on `RooFit` and provides the statistical tools for hypothesis tests, confidence intervals, and other types of inference [29]. The factorization of the modelling (i.e., `RooFit`) and the inference (i.e., `RooStats`) is important conceptually and technically, and the success and stability of the interfaces is an indication that the factorization is robust and mature.

A key innovation was the introduction of the `RooWorkspace` through the `RooStats` project, which provides both a convenient interface to `RooFit` as well as a serialization of statistical models and associated data [17, 29]. Workspaces can include multiple datasets (e.g., observed data, Asimov data, pseudo data, modeled with `RooAbsData`), statistical models (`RooAbsPdf`), and model metadata (`ModelConfig`) that identify parameters of interest (POI) $\mu$, the nuisance parameters $\theta$, the primary measurements $x$, and the auxiliary data $y$. Workspaces are currently being preserved within the LHC collaborations and form the basis of inter-collaboration combination efforts such as the *LHC Higgs Combination Group*. Such workspaces represent ready-made data products that could be published on open archives such as `HEPData`. Recommendation 3c in the 2012 *Les Houches Recommendations for the Presentation of LHC Results* [7] called for the publication of these workspaces or a similar digital implementation, and this recommendation has been restated here in Section 2.2.

In addition to the underlying library of components and the modelling language provided by `RooFit`, there are a number of specialized tools built on top of `RooFit` and `RooStats` that have been developed to aid the user in a particular context. For example, a physics group may want to build a scripting language or *factory* for building a certain class of statistical models, or they may want a high-level tool that provides an integrated workflow that includes model building and inference. Figure 2 provides a brief guide to common tools focusing on which aspects of the workflow (modelling, inference, integrated workflow) they target. By focusing on the serialization aspect (reading and writing the statistical models) and leveraging the success of the abstract interfaces used in `RooFit` and `RooStats` as a guide, we can avoid the details of these higher-level tools.

## 3.1 Infrastructure for open-world models

As mentioned in the previous section, `RooFit` allows users to extend the library of modelling components by inheriting from the `RooAbsPdf` abstract interface, which permits an open-world approach to modelling. The `RooWorkspace` is a comprehensive solution to sharing and archiving statistical models, but comes with a few drawbacks. The `RooWorkspace` design is deeply tied to its implementation in `ROOT`. While this allows capturing open-world models, it implicitly adds a heavy software dependency. To be able to load and use the serialized models, the user must have the associated software libraries that define the new modelling components[4], which, unfortunately, starts down the difficult slippery slope of generic software preservation.

Ideally, the statistical models would be published in a serialized format together with a declarative specification that provides a mathematical definition of the model.[5] This presents a challenge for an open-world approach as one must develop a declarative specification that is both readily understood and general enough to describe new modelling components.

---

[4]`RooFit` does provide the ability to store the corresponding code and build the libraries on the fly, but this functionality is not heavily used or well tested. In practice, the experiments usually maintain custom `ROOT` builds that have the necessary software patches and extensions.

[5]This echoes Recommendation 3b in the 2012 *Les Houches Recommendations for the Presentation of LHC Results* [7].

**Brief guide to commonly used statistical tools**

- `RooFit` provides a statistical modelling language and a large library of common modelling components. It also provides the `RooWorkspace`, which in turn provides a convenient interface to `RooFit` and is a container for a statistical model, data, and metadata that can be serialized using ROOT's file format and serialization technology. These workspaces are the primary tool for combining statistical models, for example in the ATLAS+CMS Higgs combinations.

- `RooStats`, which is built on top of `RooFit`, provides tools for statistical inference (hypothesis testing, confidence intervals based on asymptotic properties of the profile likelihood ratio, pseudo-data generation, credible intervals — the Bayesian analog of confidence intervals).

- `HistFactory` is a widely used tool that provides a small subset of `RooFit` modelling components and defines a declarative specification for binned statistical models in a closed-world approach. `HistFactory` focuses on the statistical modelling stage.

- `HistFitter` and `TRExFitter` are tools built on top of `HistFactory`, `RooFit`, and `RooStats` that provide an integrated workflow that includes statistical modelling and inference [30].

- `Combine` is a tool built on top of `RooFit` and `RooStats` that provides an integrated workflow and includes a scripting language to build a large variety of statistical models. It is heavily used within CMS and can produce binned statistical models similar to `HistFactory`. Due to the flexibility of `Combine`'s modelling capability it is closer to an open-world approach.

- `pyhf` is a pure-Python implementation of the `HistFactory` specification that is independent of ROOT and uses JSON as a serialization format. It is possible to convert models between the ROOT-based `HistFactory` and `pyhf`, and the `pyhf` models are now being published to `HEPData`.

- `cabinetry` is a Python-based tool similar to `HistFitter` and `TRExFitter` that can be used to build `pyhf` models and perform visualization and inference tasks.

- `zfit` is a Python-based tool similar to `RooFit` that focuses on unbinned statistical models [31]. As of this writing, `zfit` does not support serialization.

- `BAT.jl`, the Bayesian Analysis Toolkit recently rewritten in the Julia language, is a multipurpose software package for Bayesian statistical inference [32]. It is possible to interface to likelihood functions implemented in languages other than Julia.

Figure 2: Brief descriptions of the main statistical tools used in particle physics as of this writing (Sep. 2021); the first five of the listed tools are built on top of ROOT.

## 3.2 Infrastructure for closed-world models

One approach to circumvent the challenges of the open-world approach is to restrict the statistical models to a finite set of building blocks and composition operations. Through a judicious

choice of building blocks and operations, a remarkable breadth of models can still be covered. Doing so makes it feasible to develop and document a clear declarative specification of statistical models, makes it possible therefore to have different mathematically equivalent implementations of the same model, and decouples the serialization from implementation details.

An early example in this direction was `HistFactory`, which provided a mathematical specification for a flexible family of statistical models for binned data [33]. The `HistFactory` specification defines how systematic uncertainties in the normalization and shape of histograms, data-driven background components, etc., are to be handled. The original `HistFactory` implementation used a human and machine-readable `XML` file together with machine-readable `ROOT` files containing histograms that provided the information required by the declarative specification. `HistFactory` also provided libraries and executables to parse the specification in order to build the model using the `RooFit` modelling language (with some extensions). The resulting model would reside in the `RooFit` / `RooStats` ecosystem,[6] could be added to a `RooWorkspace`, and could be combined with other `RooFit` statistical models.

Subsequently, `pyhf` was developed as an independent pure-Python implementation of the `HistFactory` specification. It has been established that inference results from both the `ROOT` and `pyhf` implementations agree up to machine precision [34]. `pyhf` also permits the pruning of full statistical models in order to simplify them for subsequent statistical analysis (see, for example, Ref. [35]).

The package `pyhf` introduced a declarative specification using `JSON` that combined the information from the `XML` and the `ROOT` portion of the original schema, such that the entire statistical model and the associated data are represented in a single human and machine-readable `JSON` file. `JSON` is ubiquitous as a file format and is supported in most programming languages, including common languages used in particle physics such as C, C++, Python, Fortran, Julia, and Go. Moreover, as an ASCII document it is also a good candidate for long-term archival of data products. In addition to reducing software dependencies and improving long-term archival, the choice of `JSON` as a format also enhances functionality when it comes to reinterpretation. Recall that reinterpretations that go beyond purely parametric reinterpretation require identifying and modifying or replacing one or more components $p_{ik}(x_i|\mu,\theta)$ in Eq. (6). To do so, while being mindful of long-term archival, one can use `JSONPatch` [36] — an industry standard format (itself represented in `JSON`) to describe modifications of a `JSON` document. In the context of reinterpretation, `JSONPatch` matches well the high-level semantics of reinterpretations ("remove original signal, add new signal") such that each reinterpretation can be associated with a `JSON` patching operation. Such patches, therefore, can serve as data products unto themselves.

A statistical model and associated data, represented in a declarative specification based on `JSON`, can be used in a variety of frameworks precisely because the specification is computer language- and framework-independent. Consequently, it is possible to construct a `HistFactory` statistical model in both `ROOT` and `pyhf` from the same `JSON` file. Likewise, the framework- and computer language-independence of `JSON` allows for new future implementations of `HistFactory`.

The ATLAS Collaboration has taken a significant step in this direction by starting to publish full statistical models to `HEPData` associated with published ATLAS analyses. As of the time of writing, in mid-2021, ATLAS has published 8 full statistical models [20–27] in the `pyhf` `JSON` format, and has recommended the publication of statistical models for all LHC Run 2 supersymmetry analyses. As demonstrated in Section 4.3, these published statistical models are already being used by the theory community for new work [37–39].

Thus far the section has focused on the `HistFactory` specification and the recent devel-

---

[6] `HistFactory` is distributed with `ROOT` alongside `RooFit` and `RooStats`.



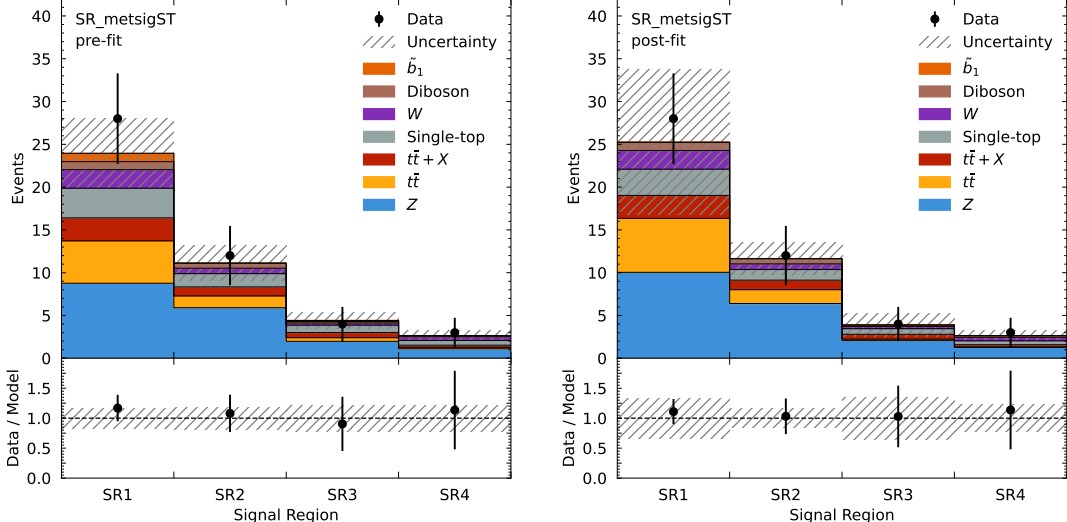

Figure 3: Pre-fit (left) and post-fit (right) visualizations of a selected signal hypothesis for four signal regions of the ATLAS search [41] of a bottom-squark of mass 600 GeV with a second-lightest neutralino of mass 280 GeV and lightest supersymmetric particle of mass 150 GeV generated from the full statistical models published in Ref. [20] using code from Ref. [40].

opments with `pyhf` and the use of the `JSON` format, which provides a template for other such efforts. We now digress to give an example of an instructive use case.

**An example user story with a published model**

To illustrate how these model serialization and analysis tools can be used today, we present an existing example [40], based on a published ATLAS statistical model, as a "user story".

A small team of physicists would like to download a statistical model from `HEPData` that was published by an LHC collaboration so that they can investigate and visualize the impacts of different model parameters on the analysis and determine how modifications of the model would impact the final results. The physicists download a `pyhf` "pallet" from `HEPData`, which contains all of the `JSON` and `JSONPatch` files that serialize the full statistical model for an ATLAS search for bottom-squark pair production [20, 41]. They are able to use the serialized model patches to build a workspace for a given signal hypothesis using `pyhf` and, then, using the `cabinetry` [42] library, investigate yields from the model. Using the same libraries, they can produce standard statistical summaries, including visualizations of pre-fit — using the initial model parameter values before inference — and post-fit distributions, as seen in Fig. 3, as well the post-fit nuisance parameter correlations useful for fit validation as seen in Fig. 4.

In addition to the full example in Ref. [40], Listing 1 demonstrates the major components of the Python APIs provided to interact with the published statistical models.

## 3.3 Approaches based on machine learning

The sections above focus on the considerations relevant for publishing the full statistical model. Here we comment briefly on other approaches.

It is possible to publish profile likelihood scans (that is, the profile likelihood $L_{\mathrm{p}}(\mu)$ at a discrete set of $\mu$ values) in many formats. For example, in Section 4.2.3 we highlight an example where ATLAS published text files tabulating the value of the profile likelihood ratio

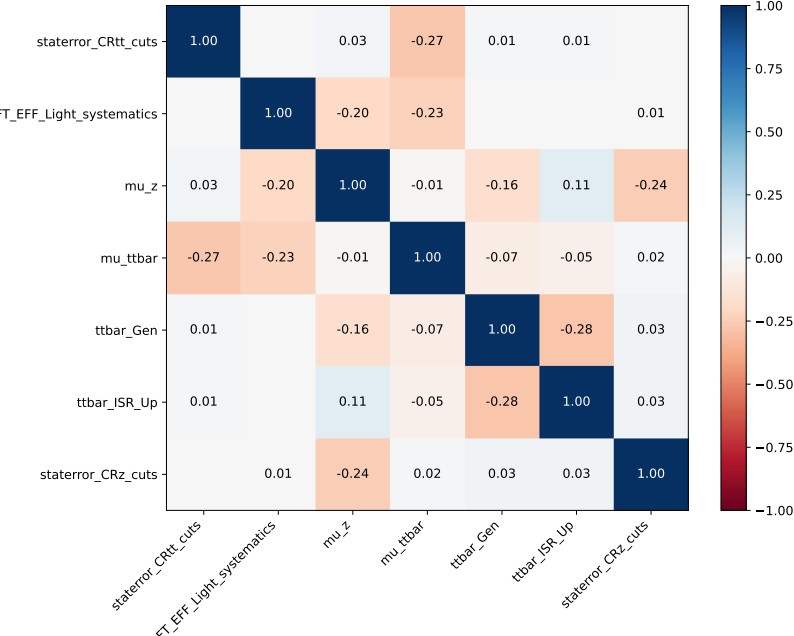

Figure 4: Post-fit visualizations of the leading nuisance parameter correlations generated from the full statistical models published in Ref. [20] using code from Ref. [40]. For better visualization, the nuisance parameter correlations shown are for the model nuisance parameters that have a correlation with any other nuisance parameter greater than 0.2 in magnitude.

in the $(\mu_{\text{ggF}+ttH}, \mu_{\text{VBF}+VH})$ plane for the Higgs boson decaying to dibosons [43–46]. However, for higher-dimensional likelihoods with nuisance parameters or several parameters of interest, this approach does not scale.

Instead, one can attempt to approximate the likelihood $L(\mu, \theta)$ with, for example, a neural network, and serialize the network using one of several technical solutions provided by the machine learning libraries (either framework-specific formats or framework-independent formats such as ONNX [47]). This is the approach taken by DNNLikelihood [48] (see also Section 4.4.4), which demonstrated that a realistic LHC-like statistical model can be encoded in neural networks with rather simple architectures with minimal loss of accuracy and trained in a reasonable amount of time.

However, without more thought it is not possible to form correctly a combined likelihood in this approach because the constraint terms associated with common sources of systematic uncertainty would be included multiple times, which has the effect of artificially reducing the uncertainty. This motivated proposals in Refs. [49] and [50] for how one might publish (profile) likelihood scans together with additional information that allows for approximate combinations. This approach requires further studies to assess the accuracy of the likelihood approximations and determine whether they are adequate for typical reinterpretation applications. The natural place to publish such "digitized" likelihoods and related documentation is HEPData.

Lastly, there are recent approaches to simulation-based inference [15, 51, 52] such as MadMiner [53–55] that use machine learning to approximate the functional dependence of $p(x|\mu, \theta)$ or $p(x|\mu, \theta)/p_{\text{ref}}(x)$ on both the data $x$ and the parameters $(\mu, \theta)$. Unlike the DNNLikelihood approach, which focuses on approximating the full statistical model built using approaches such as RooFit, HistFactory, Combine, etc., the goal of these tools is to serve as the primary statistical model when the data $x$ are high-dimensional and the traditional

```
1  import json
2  import cabinetry
3  import pyhf
4  from cabinetry.model_utils import prediction
5  from pyhf.contrib.utils import download
6
7  # download the ATLAS bottom-squarks analysis probability models from HEPData
8  download("https://www.hepdata.net/record/resource/1935437?view=true", "bottom-squarks")
9
10 # construct a workspace from a background-only model and a signal hypothesis
11 bkg_only_workspace = pyhf.Workspace(json.load(open("bottom-squarks/RegionC/BkgOnly.json")))
12 patchset = pyhf.PatchSet(json.load(open("bottom-squarks/RegionC/patchset.json")))
13 workspace = patchset.apply(bkg_only_workspace, "sbottom_600_280_150")
14
15 # construct the probability model and observations
16 model, data = cabinetry.model_utils.model_and_data(workspace)
17
18 # produce visualizations of the pre-fit model and observed data
19 prefit_model = prediction(model)
20 cabinetry.visualize.data_mc(prefit_model, data)
21
22 # fit the model to the observed data
23 fit_results = cabinetry.fit.fit(model, data)
24
25 # produce visualizations of the post-fit model and observed data
26 postfit_model = prediction(model, fit_results=fit_results)
27 cabinetry.visualize.data_mc(postfit_model, data)
```

Listing 1: Example use of the `pyhf` v0.6.3 and `cabinetry` v0.3.0 APIs for interacting with the published statistical models. This example is standalone fully runnable code, but it is only meant to highlight the major components of the statistical model use and uses `cabinetry` as it offers the highest level API to an analyst.

modelling approaches are inadequate.

Technically, these machine-learning based surrogates go beyond the likelihood function since they depend on the observable data, but, typically, it is not possible to identify the internal details of the model such as signal and background components, etc. In some cases, these neural network-based models may still allow the generation of pseudo data[7] $x \sim p(x|\mu, \theta)$, while in others the neural network approximates a likelihood ratio and does not permit the generation of such data. It should be noted, however, that nothing prevents the use of machine learning models to approximate the individual components of, for example, Eq. (7), in which case access to all components of a statistical model would be available.

### 3.4 Other related examples

**HAMMER** The `HAMMER` library [56] provides a fast and efficient algorithm to reweight large $b \to \{c, u\} \tau \bar{\nu}_\tau$ simulated samples to any desired new physics scenario generated by the relevant ten four-Fermi operators. In addition, changes of hadronic form factors to evaluate uncertainties or float such as additional nuisance parameters in a minimization problem, can be introduced. Although the code itself does not directly construct likelihoods, it provides the LHCb and Belle II experiment with the necessary key tools to present experimental data in a model-independent way — a concrete toy example of which is discussed in Section 4.4.3. The code further allows experiments to reuse their large dedicated SM MC samples for new physics

---

[7]Such techniques typically use a class of neural networks known as normalizing flows.

interpretations. The algorithm is based on event-weights of the form

$$\sum_{\alpha,i,\beta,j} c_\alpha c_\beta^\dagger F_i F_j^\dagger W_{\alpha i \beta j}, \tag{11}$$

that are proportional to the ratio of the differential rates (and thus depends on the final state kinematics). Here $c_{\alpha/\beta}$ denote Standard Model (SM) or new physics (NP) Wilson coefficients, $W_{\alpha i \beta j}$ denote a weight tensor (built from the relevant amplitudes describing a process in question), and $F_{i/j}$ encode hadronic form factors. The key realization is that the sub-sum $\sum_{ij} F_i F_j^\dagger W_{\alpha i \beta j}$ is independent of the Wilson coefficients. Once this object is computed for a specific event it can be contracted with any choice of new physics to generate efficiently an event weight. In an eventual fit, observed events often are described by binned data. This allows one to carry out the individual sub-sums and store them in histograms, which in turn can be used to produce efficient prediction functions. In Ref. [57] an interface for `RooFit` was presented, which admits an alternative usage in standalone `RooFit/HistFactory` analyses.

**Fermitools**   The Fermi Large Area Telescope (Fermi-LAT) is a space-based gamma-ray telescope launched in 2008 and operational since then. The LAT has all the basic ingredients of a particle physics detector (silicon tracker, CsI calorimeter, veto detector) [58] and mainly provides the directions and energies of the observed gamma rays. The requirement that the data and associated analysis tools be published approximately a year after the end of commissioning led to the development of `Fermitools` [59], which provide pre-defined, and allow user-defined, statistical models to be convolved with parametrized detector response functions. Different classes of event selections are offered with respective response functions corresponding to different levels of background [60]. Examples of relevance to particle physics include the search for annihilation signals from dark matter in dwarf galaxies [61–63]. Another example, where these tools have been applied by users outside of the Fermi-LAT Collaboration is the characterization of an excess of gamma rays from the center of the Milky Way in terms of dark matter (see, for example, Ref. [64]).

The approach taken by Fermi-LAT is not so much to publish likelihood functions for given models but rather to provide the community with easy to use tools to allow individual scientists to implement their own analysis. Models of universal backgrounds (isotropic, Galactic diffuse gamma-ray emission and point sources) are provided as templates (in `fits` format or as text files) [65]. Likelihoods for specific models given specific datasets are not published as part of the `Fermitools`, but some individual analyses decided to publish likelihood functions in machine-readable format (see e.g. Ref. [62]). An early example of application of `Fermitools` in a particle physics context is the use of data for dwarf galaxies to search for supersymmetric dark matter [66]. In this case the publicly available event selection, detector response functions and backgrounds were used but the convolution with detector response functions was implemented by the author for faster computation. Current implementations of BSM global fits (see also Section 4.7) use the above mentioned machine-readable likelihoods.

## 3.5   Implementation of FAIR principles

The focus and primary recommendation of this white paper is to publish statistical models as fully-public data products. With this goal in mind, it is important to ensure that these data products be available for use in a clear and consistent framework and that their locations will be stable with long-term support [67].

Best practice in this context is that statistical models and their associated data be published in a manner that adheres to the Findable, Accessible, Interoperable, and Reusable (FAIR) principles for scientific data management [68]. A specific example from the `FAIR4HEP`

project [69] of the application of FAIR principles to high energy physics (HEP) data can be found in Ref. [70], which includes a domain-agnostic, step-by-step assessment guide to evaluate the degree to which a given data product adheres to the FAIR principles. Moreover, it demonstrates how to use this guide to evaluate the FAIRness of an open, simulated dataset produced by the CMS Collaboration. Additionally, the data products published on `HEPData` are strongly guided by considerations around FAIR best practices and, therefore, offer additional working examples of FAIR implementations of publishing data products.

# 4 Physics examples, use cases

We now give concrete examples of how full likelihoods and, ideally, the availability of full statistical models can enhance the impact of experimental results.

## 4.1 Parton distribution functions

The quark and gluon substructure of the proton, quantified by the parton distribution functions (PDFs) [71, 72], plays a crucial role in the interpretation of most LHC analyses. On the one hand, the PDF uncertainties are an important (correlated) source of theory errors for Higgs boson characterization [73], searches for new particles at high mass [74], and the measurement of fundamental SM parameters from $m_W$ [75] to the strong coupling parameter $\alpha_s$ [76]. On the other hand, LHC data themselves provide an ever-growing set of constraints on the proton PDFs [77], and hence modern global fits [78–80] include a wide range of data from LHC processes such as inclusive $W$ and $Z$ production, single-jet and dijet cross sections, and top-quark pair production.

Unfortunately, fully exploiting the information provided by the LHC on the proton PDFs is often hampered by limitations in the publicly available statistical information provided by the experimental collaborations. Global PDF fits are based on the minimization of the $\chi^2$,

$$\chi^2 = \frac{1}{n_{\text{dat}}} \sum_{ij} \left( \mathcal{F}_i^{(\text{exp})} - \mathcal{F}_i^{(\text{th})}(\boldsymbol{\mu}) \right) \left( \text{cov}^{-1} \right)_{ij} \left( \mathcal{F}_j^{(\text{exp})} - \mathcal{F}_j^{(\text{th})}(\boldsymbol{\mu}) \right), \tag{12}$$

which compares the theory predictions $\mathcal{F}_j^{(\text{th})}(\boldsymbol{\mu})$ expressed in terms of the PDF model parameters $\boldsymbol{\mu}$ to the experimental data $\mathcal{F}_i^{(\text{exp})}$ by means of the covariance matrix. Note that Eq. (12) assumes that the data are Gaussian distributed around the true (SM) values, which may or may not be a good approximation. Ideally, the full details of the $n_{\text{dat}}$ bin-by-bin correlated errors, both statistical and systematic, are provided in which case the covariance matrix can be schematically constructed as

$$\text{cov}_{ij} = \delta_{ij} \sigma_i^{(\text{stat})} \sigma_j^{(\text{stat})} + \sum_{k=1}^{n_{\text{sys}}} \sigma_i^{(\text{sys})(k)} \sigma_j^{(\text{sys})(k)}, \qquad i, j = 1, \ldots, n_{\text{dat}}. \tag{13}$$

However, in practice, the relevant information to construct Eq. (13) and hence Eq. (12) is often either unavailable or provided in a form that renders its usage impossible or cumbersome. Most of the situations that one encounters when interpreting LHC data in the context of global PDF fits fall in one of the following categories:

   i Only the total systematic error is provided and information about correlations is missing. Then, there is little choice but to add the systematic and statistical errors in quadrature, which reduces the PDF sensitivity of the measurement.

ii  Only the full covariance matrix is provided, without details of the individual components. This complicates the PDF interpretation of the results because, e.g., such covariance matrices need to be modified to avoid the D'Agostini bias [81, 82]. Furthermore, in case of ill-defined covariance matrices (see below) one cannot identify the relevant sources of systematic errors causing the problem.

iii  The provided covariance matrix is not positive-definite (e.g., it has some negative eigenvalues) or it is ill defined. In this case, the dataset is either discarded or its covariance matrix regularized by some *ad hoc* procedure.

iv  The covariance matrix turns out to be sensitive to subtle differences in the assumptions concerning its correlation model. In these cases, the $\chi^2$ in Eq. (12) can deviate markedly from unity even when theory and data are fully compatible. This situation would likely be avoided if the "error on the errors", or sometimes more importantly the "error on the correlations" were to be accounted for.

It is important to emphasize that these problems are becoming more acute with the availability of high-precision LHC measurements, which are more often than not limited by systematic rather than by statistical uncertainties. Furthermore, these issues also affect the eventual combination of datasets from a given experiment, which share common systematic error sources, within the global PDF fit. Ensuring that new LHC measurements are always accompanied with the release of the full statistical model of the measurement (and associated data) would solve or at least alleviate some of the problems that currently hamper the PDF interpretation of LHC data. Specifically, (i) to (iii) would certainly become obsolete, while in the case of (iv) the relevant information would become available to construct alternative correlation models — a procedure now sometimes attempted by the PDF fitters — and study the robustness of the resultant PDF fits with respect to them. We note that progress in this direction has been made for ATLAS 8 TeV inclusive jet cross sections [83].

The availability of statistical models, or at least full likelihoods, would also facilitate the inclusion in global PDF fits of analyses not considered so far, such as high-mass searches in the dilepton final state at the detector level [84] which provide interesting constraints on the large-$x$ PDFs. An example of a low-statistics dataset which in principle provides information required for PDF fits may be found for high-$x$ ZEUS structure function data [85]. Similar techniques could be applied to low-statistics data in tails of high-energy LHC datasets, if the full statistical models were provided.

Some recent examples of category (iv) include issues related to the interpretation of differential top-quark pair production data in terms of the gluon PDF [86–88] and the analysis of inclusive jet and dijet cross sections [89, 90], where decorrelation or regularization approaches are required to achieve a sensible PDF fit. Furthermore, one should note that possible tensions observed between datasets might not have an underlying physical origin but may arise instead from the limited available experimental information or affected by the sensitivity to the correlation model described above. This could have a significant impact on the so-called "tolerance", or artificially inflated $\Delta\chi^2$ introduced in some global PDF fits, e.g., Ref. [78, 79] in order to account for the tension between different datasets. An example of a dataset which is a source of tension in PDF fits is the ATLAS $W, Z$ 2011 dataset, a high-precision measurement which is omitted from the CT18 global fits because of potential tensions with other datasets, which are absent when a dataset is interpreted independently of others [91].

Some of these sources of tension may arise from the theoretical uncertainties in the PDFs due to the limits of perturbative QCD calculations. We note the first attempts to understand theoretical uncertainties related to the PDF extractions [92, 93], and in theoretical perturbative calculations, e.g., Ref. [94]. Ideally, in the future it will be possible to present the full

PDF uncertainties from both experimental and theoretical sources along with an appropriate procedure for applying them in predictions and in modelling.

To summarize, a key message to emphasize here is that the availability of open statistical models would bring research in proton structure, as well as in closely related fields from the determination of nuclear PDFs [95, 96] to that of hadron fragmentation functions [97], to a whole new dimension. In particular, the PDF community could finally move beyond the multivariate Gaussian approximation, and include the tails of LHC distributions which contain valuable PDF-sensitive information on, e.g., the poorly known large-$x$ gluons and sea-quark PDFs. Such tails are now routinely discarded in "SM measurements" since the Gaussian approximation does not hold for such bins. This is serious limitation. Open statistical models would also allow a much closer study of the possible interplay between PDF fits in BSM searches in high-$p_T$ tails. An inability to disentangle effects of PDFs from those of new physics could be a showstopper for the HL-LHC program [98, 99].

Furthermore, as illustrated by some of the explicit cases above, global PDF fits start to be limited by tensions and inconsistencies that are almost unavoidable when statistical errors become negligible. The usual covariance matrix approach is simply not suitable to tackle this problem, and the release of the full likelihood (ideally encapsulated in the statistical model) is the only option to fully exploit the information contained in the next generation of precision LHC measurements. Without this sea change in practice, we may end up swamped by spurious tensions between data and theory models (either SM or BSM) that merely reflect the inappropriate use of a Gaussian error model in PDF fits to experimental measurements.

## 4.2  Higgs boson measurements and EFT interpretations

### 4.2.1  Theorists' perspective

Global interpretations of LHC data in terms of EFT parameters [100–104] are affected by many of the issues described in the PDF case in Section 4.1. This cannot be otherwise, given that an EFT fit based on "SM measurements" proceeds usually by means of the same $\chi^2$ minimization as in Eq. (12), where now the model parameters $\boldsymbol{\mu} = \boldsymbol{c}/\Lambda^2$ are the Wilson coefficients associated with the EFT operators. Working at dimension-six in the EFT and retaining both the interference and quadratic terms, a generic LHC cross section will be modified as follows

$$\sigma_{\text{LHC}} = \sigma_{\text{SM}} \left( 1 + \sum_i \frac{c_i}{\Lambda^2} \kappa_i + \sum_{i,j} \frac{c_i c_j}{\Lambda^4} \widetilde{\kappa}_{ij} \right), \tag{14}$$

where the sum extends over all operators that contribute to the process under consideration for a given set of assumptions, e.g., concerning the flavor structure of BSM physics. For the data relevant in EFT analyses, in contrast to the typical measurements used in PDF fits, one often ends up in categories (i), (ii), and (iii) above, which hampers the statistical interpretation of LHC measurements in terms of bounds on Wilson coefficients. Clearly, as the data used in EFT fits become dominated by systematic effects and uncertainties, we may end up in the same situation as for PDF fits, namely, being unable to determine whether finding non-zero EFT coefficients are hints of BSM physics or are artefacts arising from limitations in theoretical calculations or the publicly available experimental information.

It is worth pointing out that some EFT analysis frameworks go beyond Eq. (12) and determine the Wilson coefficients directly from a likelihood maximization procedure that does not rely on the Gaussian approximation. Such an approach has two main advantages. Firstly, it provides a principled way to include theoretical uncertainties in the error budget, and to treat them using non-Gaussian (e.g., flat) prior distributions. Secondly, it permits inclusion of "search" data provided at the detector (folded) level, which represent the bulk of the information used as input in PDF fits, rather than unfolded data. The search data are particularly

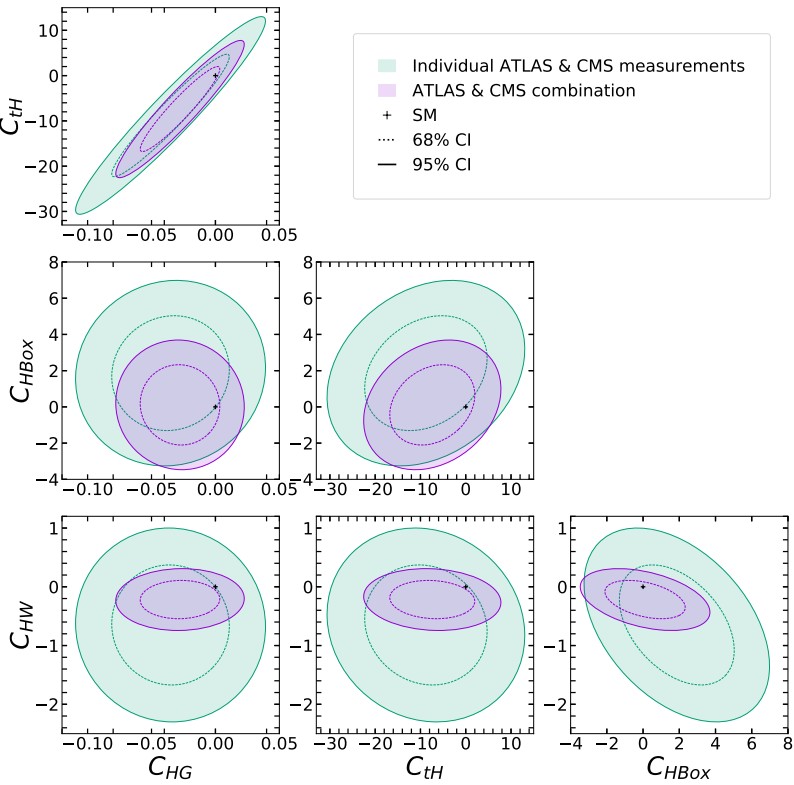

Figure 5: Constraints on operators of the dimension-6 SMEFT using data from the combination of Run 1 Higgs data performed by ATLAS and CMS, and using data from individual ATLAS and CMS Run 1 measurements. Shown are the marginalized constraints on two operators at a time; dashed (solid) lines correspond to 68 (95)% CI.

useful for EFT fits since they include high-$p_T$ regions that exhibit high sensitivity to the Wilson coefficients; moreover, many distributions are measured that help close open directions in the EFT parameter space.

However, given that full likelihoods are not yet routinely available, the latter approach requires a number of approximations from the theory analysts (see also Section 4.3) that may compromize the reliability of the results. Again, the availability of the full statistical model together with the associated data would render this problem moot, since one can then construct the exact likelihood function without the need of any *ad hoc* approximation and thus exploit the complete information offered by the LHC data.

This point can be illustrated by comparing the sensitivity to EFT operators from LHC Higgs boson measurements. In particular, we constrain five coefficients of the dimension-6 SMEFT[8] ($C_{HW}$, $C_{tH}$, $C_{HG}$, $C_{HBox}$ and $C_{HB}$) using LHC Run 1 Higgs boson data, comparing the CMS and ATLAS combination [105] with the individual ATLAS and CMS Higgs boson results [106, 107]. The former is a combination performed by the experimental collaborations themselves in which information about the full statistical model was available and correlations between the analyses could be taken into account (see Section 4.2.2). Concretely, we take a set of 23 signal strengths representing the best available measurements from Table 8 of Ref. [105], along with the correlation matrix also provided in [105]. The data input to our fit consist of

---

[8]See Ref. [102] for notation and conventions.

central values and their asymmetric uncertainties along with the correlation matrix, where we make use of the "Variable Gaussian (I)" likelihood of Ref. [108] (see also Ref. [109]). In contrast, when combining the individual ATLAS and CMS measurements [106, 107] we assume zero correlation between the analyses. The data from these individual papers are presented in the $\sigma \times$ BR parametrization consisting of cross sections, branching ratios and their ratios. Correlation information is provided for the ATLAS measurements but not for the CMS measurements, and is therefore neglected in the latter case.

Figure 5 shows the 68% and 95% confidence intervals (CI) on the dimension-6 SMEFT co­efficients $C_{HW}$, $C_{tH}$, $C_{HG}$ and $C_{HBox}$, taking two operators at a time while marginalizing over the unseen directions. We also marginalize over $C_{HB}$, but do not show its constraints here as it is fully anticorrelated with $C_{HW}$. Comparing the purple and green areas, we find shifts in the central value of each coefficient, with the fit to individual ATLAS and CMS measure­ments resulting in weaker constraints. These results highlight that some information is lost in the combination without correlation information. Comparing the green and purple regions demonstrates how these constraints can be improved when information from the likelihood is available: by recombining measurements into a signal strength parametrization, higher sensi­tivity to EFT operators is achieved. This comparison was produced using the `Fitmaker` public code [102, 110].

Higgs boson measurements in terms of signal rates and/or simplified template cross sec­tions (STXS) are also routinely used to constrain *explicit* models of new physics through their effects on the Higgs boson couplings, and two public codes exist for this purpose: `HiggsSignals` [111, 112] and `Lilith` [109]. Again, the construction of an approximate likelihood from the published experimental results is made difficult by limited or missing in­formation on i) the shapes of uncertainties and ii) the channel-by-channel correlations. For LHC Run 2 results, this is discussed in detail in Refs. [109, 112].

A showcase of the typical challenge of such reconstructions of the likelihood is presented in Fig. 6 (taken from Ref. [112]; additional examples are discussed in Ref. [109]). Such plots are not an interesting physics result in themselves, but they are a stringent test of the implemen­tation of the approximated likelihood and they expose different possible pitfalls. The top row shows the comparison of the constraints on the signal strength modifiers $\mu_{ggF}$ and $\mu_{VBF}$ for a CMS $h \rightarrow WW$ search [113, 114] from the early LHC Run 2. In the left panel, no efficiencies for the measured signal rates in the individual kinematic bins of the rate measurements were given. In this case, the efficiencies need to be either evaluated with experiment-independent simulation tools, taken from truth-level analyses, or estimated from earlier, similar analyses. As the left panel shows, an error in the assumed efficiencies easily leads to a wrong likeli­hood. As the right panel shows, this problem is easily averted when the missing information is included. This challenge in itself is easy to avoid. Note, however, that the publication of full likelihoods could still contain such challenges if the exact specification of the model in the likelihood is unclear. The publication of likelihoods does not relieve one from the need to choose an effective parametrization of the likelihood using parameters that can be derived in all applicable models, and a sufficiently complete statement of all assumptions that underpin the statistical model.

The middle row shows the STXS combination of ATLAS SM Higgs boson measurements with 80 fb$^{-1}$ of LHC Run 2 data. The validation is performed in the $\kappa$ framework [117]. In the left panel, the provided correlation matrix of the uncertainties of the STXS bins is consciously omitted, in the right panel it is applied. This simple example highlights the importance of publishing details of the fit model, which would automatically be included in the full likelihood.

The bottom row shows an example from the validation of the ATLAS $h \rightarrow 4\ell$ STXS measure­ment implementation in `HiggsSignals`. In the left panel, the $\kappa$ framework results are shown without the application of correlated signal theory systematic uncertainties, while the right

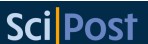

**Figure 6:** Constraints on Higgs boson production and couplings derived in [112] based on Higgs boson rate measurements in the $H \to WW$ decay from CMS [113], [114] *(top row)*; on combined STXS measurements from ATLAS [115] *(middle row)*; and on the $H \to 4\ell$ decay from ATLAS [116] *(bottom row)*. In the top and middle rows, improving the input to the approximate likelihood reconstruction *(right panels)* improves the deficiencies seen when necessary information is omitted *(left panels)*. In the bottom row, even the most complete available input still yields deviations between the ATLAS results and the reconstruction of the likelihood, which cannot be described by the publication of results in formats like STXS cross section values and uncertainties. Instead, a more complete representation of the likelihood is required. See text for details.

panel includes these. For the published likelihoods, this example highlights the importance of the access to the signal and background theory uncertainty model. In addition, this low-statistics but high-precision channel also shows the existence of fundamentally non-Gaussian uncertainties, which cannot be reconstructed from the (asymmetric) published uncertainties on the STXS results. These become more and more apparent as one moves further and further away from the minimum of $-2\Delta\ln\mathcal{L} \approx \Delta\chi^2$. The publication of the full likelihood would naturally include all non-Gaussian features.

Other examples of Higgs boson measurements where a more complete likelihood model would be important include, e.g., the $h \to \tau\tau$ based measurement of the Higgs boson CP by CMS [118]. While a profiled likelihood is given as a function of the CP phase $\Phi_{CP}$ and the signal rate modifier $\mu$, the sensitivity of the constraints depends critically on the relative contribution of the VBF vs gluon-fusion cross sections, which is not given. In this case, a likelihood as a function of the *relevant* physical dimensions would already be very useful, as long as the full statistical model is not yet available. Similar challenges also appear in the treatment of other Higgs boson CP constraints, see e.g. Ref. [119].

### 4.2.2 Perspective from "official" experimental combinations

Within the ATLAS and CMS experimental collaborations, combined Higgs boson measurements have been produced using data from Run 1 of the LHC [105,120]. The results published are *not* combinations of the individual measurements performed by each experiment, but rather a set of *combined measurements* of Higgs boson properties using simultaneously the Run 1 datasets collected by ATLAS and CMS. While the two experimental collaborations use different software to construct likelihood functions and perform statistical inference, both are based on the `RooFit` and `RooStats` packages and the serialization of the data and the statistical model $p(x|\mu,\theta)$ uses a common container format provided by the `ROOT` based `RooFit` package: the `RooWorkspace` [17,29]. This allows the exchange of the full statistical model (as in Eq. (7)) and data between the experiments, which makes it possible to reconstruct and aggregate the individual terms for each experiment for the observed data (or pseudo data) and perform statistical inference[9]. In particular, since the full parametrization of the statistical model is provided, nuisance parameters can be correlated without duplicating the constraint terms associated with common sources of systematic uncertainty, the constraint terms can be updated, and re-parametrization of the statistical model is relatively straightforward.[10]

The way that ATLAS and CMS share their full statistical models and perform combined measurements (that is, global inferences) is an excellent example of the gold standard that we advocate. These collaborations share workspaces that include both closed-world models, which are straightforward to work with, as well as open-world models that are more difficult to work with because of their software dependencies.

Currently, neither the individual nor the combined `RooWorkspaces` are published alongside the measurements by either collaboration for the Higgs boson combined measurements. This is largely due to historical practice, the perceived difficulty of using the information by non-experts (where "expert" means expert in the analyses entering the combination), and the relatively large computational resources (both in terms of memory and CPU time) involved in performing statistical inference using combined Higgs boson likelihoods. These concerns

---

[9]Such likelihood-based combinations have been the gold standard for combined Higgs boson searches since the time of the 2000 CERN workshop on confidence limits [1]. This approach was used by both the LEP and Tevatron Higgs boson working groups [121,122]. Prior to the advent of the `RooWorkspace`, however, the combination was technically cumbersome and less flexible.

[10]The exception is modifications of the statistical models that require information not already included in the model components, such as updating the simulation to use a new MC generator. This is particularly relevant for EFT and BSM interpretations where one may need to recalculate acceptance/efficiencies for signal and background components that are needed to construct $p_i(x_{ij}|\mu,\theta)$.

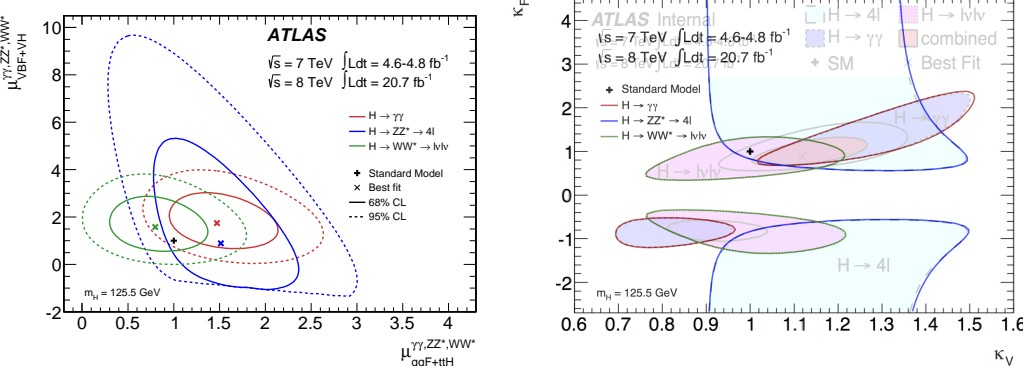

Figure 7: Left: Figure 7 from Ref. [43]. The corresponding 2-D profile likelihood scans are published on HEPData [44–46]. Right: An overlay of Figure 10 from Ref. [43] and the reproduction of the confidence interval contour based on the published likelihoods. Note the non-Gaussian nature of both sets of contours, and the accuracy of the reproduced results (the colored lines of the reproduced contours overlap with the boundary of the original shaded regions).

should be taken seriously. However, they can be factorized from the considerations about whether or not to publish the statistical models themselves and addressed independently. We have seen recently a marked shift in attitudes as the tools have become easier to use, more robust, and the value of these data products is more widely recognized. In particular, the ATLAS Collaboration has started to publish `HistFactory` models saved in `pyhf`'s human-readable format (see Section 3.2) instead of the `RooWorkspace` format. We note, however, that the format choice for publishing models is a separate issue from the decision about whether or not to make the statistical models public.

### 4.2.3 An example of likelihood publishing for Higgs boson characterization

While the collaborations have not released the statistical models of any Higgs boson measurements (yet), there are examples of publishing 1-D and 2-D profile likelihood scans $L_p(\mu)$. For example, ATLAS published simple text files tabulating the value of the profile likelihood ratio in the $(\mu_{ggF+ttH}^f, \mu_{VBF+VH}^f)$ plane for the Higgs boson decaying to dibosons [44–46]. Figure 7 (left) shows the 68% and 95% confidence level contours, which reveal departures from Gaussianity. Importantly, this information is enough to derive confidence intervals in a different parametrization based on the $\kappa_F$–$\kappa_V$ framework that preceded a full EFT analysis. Figure 7 (right) shows confidence intervals in those models, which have complex shapes that are not well approximated by an elliptical contour. Furthermore, the solid colored curves were produced from the published likelihoods and overlayed on top of the original figure published by ATLAS. This shows excellent agreement in each of the individual channels.

However, it is not possible with this information to reproduce the combined $\kappa_F$–$\kappa_V$ confidence interval because the constraint terms associated with common sources of systematic effects would be double- or triple-counted and the common nuisance parameters would be independently profiled to inconsistent values. This motivated a proposal in Ref. [49] for how to publish profile likelihood scans together with additional information that allows for such combinations, but that procedure is still approximate. To accurately reproduce the combined likelihood contour requires accessing the individual terms from each channel, which is possible with the full statistical model as described in Section 4.2.2 but not otherwise.

### 4.3 Direct searches for new physics

Reinterpretations of BSM searches aim at obtaining a global picture of the constraints that LHC results impose on specific theoretical models of new physics, and thereby go beyond the picture provided by simplified models and/or scenarios, which have generally been used by the collaborations to communicate and compare the reaches of individual analyses.

#### 4.3.1 Supersymmetry and exotics searches

The current practice in quoting ATLAS and CMS search results is to provide observed counts and background estimates with their uncertainties for the analyses' signal regions (SRs). Independent of whether the reinterpretation is done by reproducing one or more experimental analyses in MC event simulation or by using simplified-model efficiencies,[11] in the absence of the full statistical model definitions, the count information above can be used to perform a statistical analysis based (only) on a simplified likelihood (SL) approach. This approach approximates the likelihood for signal counts with Poisson distributions while assuming Gaussian likelihoods for all other quantities. If the analysis has several SRs or if multiple analyses are being combined and no details on the background correlations are available only the most sensitive (colloquially called the "best") SR is used to obtain a limit.[12] Sometimes bin-to-bin background correlations are provided, in which case it is possible to combine multiple SRs, though still in an SL approach that assumes (symmetric) Gaussian uncertainties [123]. This is clearly a step in the right direction; however, it is not always a good approximation, in particular, when the reported uncertainties are asymmetric. A simple workaround to approximate non-Gaussian uncertainties is to add a skew term in the SL [50].

While Refs. [50,123] provide useful frameworks, reinterpretation statements would clearly be of greater scientific value if they were based on the full statistical models. This point is illustrated in Fig. 8. Here we compare the 95% $\mathrm{CL}_s$ limits obtained with SModelS [39, 124] using either the best SR only, or combining SRs by means of the statistical models provided by ATLAS.[13] The examples shown are for the ATLAS multi-$b$ sbottom search [41], the stau search [125] and the electroweak-ino search in the 1-lepton+Higgs boson channel [126]. In all three cases, the improvement with respect to the best-SR limit is obvious; the residual difference between the red and black full lines comes from using (interpolated) simplified-model efficiencies to determine the signal yields rather than those from the full ATLAS simulation.

#### 4.3.2 Searches for additional Higgs bosons

An example of the publication of profiled likelihoods and a showcase for their usefulness is the search for heavy 2HDM/MSSM like Higgs bosons in the $H/A \to \tau\tau$ decay by ATLAS [127] and CMS [128]. The collaborations provided the observed and expected values of $-2\Delta \ln \mathcal{L}$ as a 3-dimensional function of the mass $m_{H/A}$ and the cross sections $\sigma_{bbH/A}$ and $\sigma_{ggH/A}$ in the $b$-quark associated production and the gluon-fusion production channels. Without knowledge of these likelihoods, there was a two-fold challenge in using the published result of this search. First, as in the other cases discussed in this section, the publication of 95 % $\mathrm{CL}_s$ limits alone is only marginally useful for global fits or combinations of searches. Second, the sensitiv-

---

[11]Simplified-model efficiencies are binned acceptance×efficiency values for particular simplified signal hypotheses.

[12]To avoid impacting the rate of false exclusions, the choice of SR per parameter point must not depend on the observed data, which are subject to fluctuations. The common approach is, therefore, to select the SR with the best *expected* sensitivity to the BSM parameter point assuming that the background-only model is true—the "most sensitive" or "best" SR.

[13]Note that for such reinterpretation purposes the published background-only JSON files are used, not those for particular signal hypotheses.

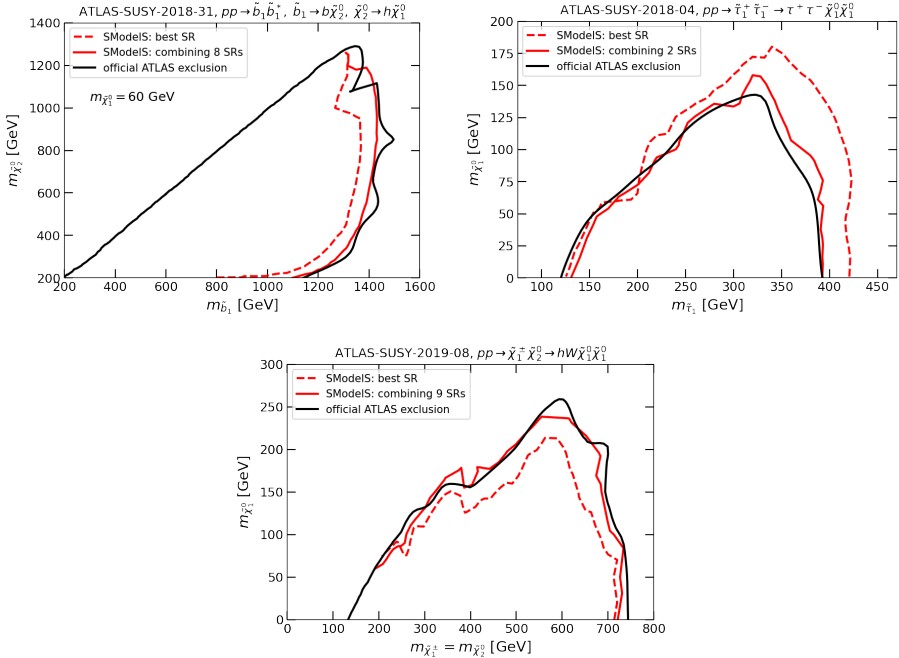

Figure 8: 95% CL$_s$ limits obtained with SModelS [39, 124] for three ATLAS analyses using the best SR only (dashed red lines) or combining SRs by means of the background-only statistical models provided by ATLAS [20, 21, 23] (full red lines); to be compared to the official exclusion lines from ATLAS shown in black.

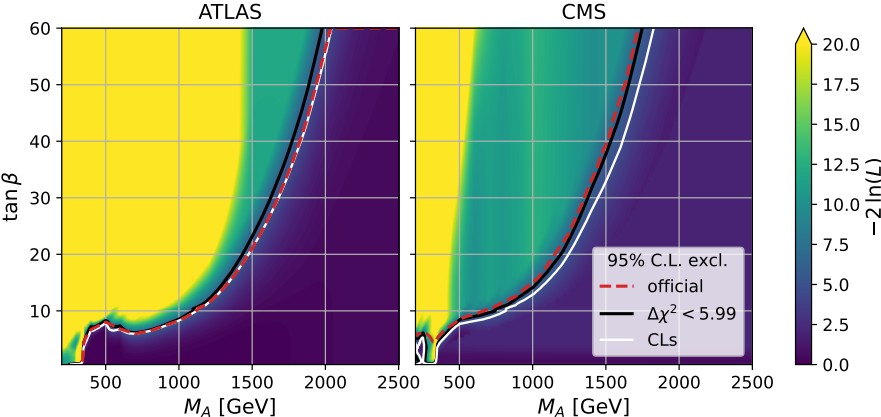

Figure 9: The reconstruction of the CL$_s$ exclusion from the published profiled likelihood in three independent dimensions of $m_{H/A}$, $\sigma_{bbH/A}$, and $\sigma_{ggH/A}$ derived in [129], based on results from ATLAS [127] and CMS [128].

ity of the $H/A \to \tau\tau$ search critically depends on the relative contribution of $b$-associated production, which has a different acceptance than the gluon-fusion channel. Since many 2HDM/(N)MSSM-like models tend to change significantly the coupling of the heavy Higgs bosons to the down-type quarks, this difference in sensitivity cannot be ignored if accurate conclusions are to be drawn.

From the published profile likelihoods, the likelihood for any model predicting neutral heavy scalars decaying into $\tau$ leptons, and being produced from either gluon-fusion or $b$-associated production in any relative contribution, can be used in any application. This also

allows the calculation of $CL_s$ values, which can either be used to derive limits on as yet unexplored models of new physics, or allows the validation of likelihood implementations by comparing limits derived from them with published results. One such example is shown in Fig. 9 [129], where the color background shows the values of $-2\Delta\ln\mathcal{L}$ in the $M_h^{125}$ benchmark model. This likelihood itself cannot be derived from the published 95 % $CL_s$ limit. For validation purposes, the $CL_s$ calculated from the published likelihood (white lines) is compared with the published limit (red dashed lines). For the ATLAS result, the two coincide perfectly, since neither the published likelihoods nor the ATLAS 95 % $CL_s$ limit in the $M_h^{125}$ benchmark model include signal cross section theory uncertainties. In the case of the CMS limit, as expected, the limit calculated from the likelihood — deliberately calculated without the inclusion of additional signal cross section theory uncertainties[14] — is stronger than the published model limit including signal theory uncertainties.

There is an interesting potential pitfall in the use of published profiled likelihoods parametrized using a limited set of physical parameters. Since the difference

$$
\begin{aligned}
-2\Delta\ln\mathcal{L}(\sigma_{bbH/A},\sigma_{ggH/A};m_{H/A}) \quad = \quad & -2\ln\mathcal{L}(\sigma_{bbH/A},\sigma_{ggH/A},\hat{\hat{\theta}};m_{H/A}) \\
& +2\ln\mathcal{L}(\hat{\sigma}_{bbH/A},\hat{\sigma}_{ggH/A},\hat{\theta};m_{H/A}),
\end{aligned}
$$

of the profiled likelihood is published in *independent* bins of $m_{H/A}$. This leads to an $m_{H/A}$-dependent shift of the 2-parameter profile likelihood maps that precludes constructing the 3-parameter profile likelihood. In particular, the maximum likelihood estimates $\hat{\sigma}_{bbH/A}$, $\hat{\sigma}_{ggH/A}$, and $\hat{\theta}$ are actually conditional maximum likelihood estimates that depend on $m_{H/A}$. In this particular example, the point $(m_{H/A}, 0, 0)$ yields the same likelihood for all $m_{H/A}$ (since no signal is present) and hence the $m_{H/A}$-dependent shift of the 2-parameter profile likelihoods can be reversed engineered, and the 3-parameter profile liklihood can be constructed. However, this would not be possible in general.

### 4.3.3 Going further: combination of analyses, global p-value, etc.

While for the above examples an appropriate modelling of the final likelihood function is sufficient, knowledge of the full statistical model allows one to go much further. Consider the example of the ATLAS multi-$b$ sbottom search [41]. The simplified model assumed in this analysis (and used in Fig. 8) is actually unrealistic in the sense that if $\tilde{b} \to b\tilde{\chi}_2^0$, $\tilde{\chi}_2^0 \to h\tilde{\chi}_1^0$ exists, there will also be a chargino near in mass to either the $\tilde{\chi}_2^0$ or $\tilde{\chi}_1^0$. Assuming a mostly bino-like $\tilde{\chi}_1^0$, we will have $m_{\tilde{\chi}_1^\pm} \simeq m_{\tilde{\chi}_1^0}$ and, therefore, the decay chain $\tilde{b} \to t\tilde{\chi}_1^-$, $\tilde{\chi}_1^- \to W^-\tilde{\chi}_1^0$ competing with the decay via the $\tilde{\chi}_2^0$. The reach for sbottoms and electroweak-inos can, therefore, be determined only if the analyses looking for these final states (including the mixed ones, i.e., $2b2h$, $2b4W$, $2b2W1h$) can be combined. Moreover, corroborating information from electroweak chargino-neutralino searches should be folded in as well. Knowledge of shared uncertainties in different analyses would permit a more global approach that would make it possible to combine analyses, albeit in an approximate way.[15]

Even within the experimental collaborations, obtaining a global view of how the various searches in individual final states constrain specific new physics models is difficult. In Run 1, ATLAS and CMS performed studies [134, 135] that interpreted a variety of supersymmetry searches in terms of the "phenomenological" minimal supersymmetric standard model

---

[14]This is done using an additional nuisance parameter in order to demonstrate the dependence on additional uncertainties between implementation-independent parametrizations (in pseudo observables such as masses and rates) and specific model-based implementations.

[15]Also interpretations like [130–133] of the tantalizing (because they are potentially consistent) small excesses in searches for additional Higgs bosons rely on combining information from different analyses.

(pMSSM) with 19 free parameters. Both studies were performed *a posteriori* after the publication of the analyses considered in the studies. The ATLAS study [134] did not try to combine the analyses and quoted only the exclusion status provided by the analysis with the best sensitivity—which is also what is done in most theoretical studies. The CMS study [135] attempted a combination of the non-overlapping subset of analyses, but did not consider cross-analysis correlations of systematic uncertainties, or those arising from control regions used in background estimation. The systematic construction, preservation, and deployment of full statistical models would have made these important studies and studies of this kind considerably more insightful, both within and outside the experimental collaborations.

Besides global top-down BSM fits, which will be discussed in Section 4.7, a reliable treatment of correlated uncertainties (including cross-analyses correlations) is particularly relevant when looking for possible dispersed signals—i.e., small effects of new physics that appear simultaneously in different final states and/or signal regions—and for a bottom-up model inference from the data as attempted in Ref. [136].

So far, lacking more detailed information, combinations of LHC searches have been done in a binary approximation: analyses are considered either approximately uncorrelated and, therefore, trivial to combine, or as correlated and, therefore, not to be combined (see, for example, Refs. [135–137]). Attempts to determine from event simulation or from analysis feature space comparisons whether analyses are approximately uncorrelated have been suggested in Ref. [37] (contributions 16 and 17). But this remains a crude treatment, which could be much improved, if not rendered obsolete, if the full statistical models were available to assess the various error sources. This would allow for the construction of more realistic combined likelihoods, as discussed in the previous sections.

The full statistical models are also needed in order to estimate global $p$-values associated, for example, with the Standard Model hypothesis in view of the BSM search results. An attempt to do this was made in Ref. [136] based on an entire database of simplified model results from nearly 100 ATLAS and CMS searches. Here, the statistical models of the analyses were sampled treating every signal region in every analysis as if it were independent of the others. This approximation holds only in the case where the experimental uncertainties are dominated by statistical rather than by systematic effects. With full statistical models for all results at hand, in which it is possible to identify parameters that have the same meaning across analyses, a single global model could be constructed, which could then be used to generate pseudo data for all analyses simultaneously. Such a sample of simulated data would take inter-analysis correlations into account in a realistic manner and thus lead to a more robust and reliable determination of the distribution of test statistics from which $p$-values and other hypothesis-testing measures could be approximated.

## 4.4 Heavy flavor physics

Heavy flavor physics focuses on the measurement of properties and decays of a $\tau$ lepton or hadrons containing heavy quarks such as $b$ or $c$. A considerable body of work in flavor physics over the past decade, both experimental and theoretical, has yielded several persistent deviations of observations from SM predictions, in particular, in the semileptonic decays of $B$ mesons. These deviations have been growing over time both in statistical significance and in global consistency. It is not surprising, therefore, that this has inspired a lot of effort in the theory community to understand and interpret these data in terms of possible new physics. Given the potential for significant advances in particle physics, should it turn out that these deviations are indeed evidence of new physics, it is of the utmost importance that all the required information and, in particular, the full statistical models and likelihood functions be published so that the experimental results can be correctly (re)interpreted.

In flavor physics, results are usually published in different forms: upper limits, single mea-

surements with symmetric or asymmetric errors, multiple measurements with symmetric or asymmetric errors, or one- or higher-dimensional likelihood functions. The reinterpretation of such measurements falls into two categories: the first category are results that have no intrinsic or a trivial dependence on underlying experimental or theoretical quantities. These can safely be analyzed to search for new physics or measure fundamental parameters of the SM, such as CKM matrix elements or other couplings. The second category of measurements exhibits a strong dependence on underlying quantities, and a direct reinterpretation would introduce an undesired bias. Examples for such are measurements that rely on, e.g., shapes of SM distributions or a specific model for signal and backgrounds. Consequently, for a consistent analysis the full underlying dependence would need to be parametrized.

### 4.4.1 Rare decays and Lepton Flavor Universality Violation ratios

Flavor physics papers typically quote a measurement as an estimate with an uncertainty. In the Gaussian limit, this provides sufficient information for a reinterpretation, as long as the measurement does not depend on the hypothesis under study. Most recent measurements of rare decays, however, handle low-event counts or incorporate systematic effects that cause the log-likelihood functions to deviate from the Gaussian limit. Consider a concrete example in the form of Lepton Flavor Universality Violation ratios: $R_{K^{(*)}} = \mathcal{B}(B \to K^{(*)}\mu\mu)/\mathcal{B}(B \to K^{(*)}ee)$. In the kinematic region, where the lepton masses can be neglected, $R_{K^{(*)}} = 1$ with a 1% uncertainty (see, e.g., Ref. [138]). Due to trigger and reconstruction inefficiencies, the LHCb experiment observed more decays with electrons in the final state as compared to muons. The number of $B \to Kee$ is so small that even assuming that the uncertainty in the number of these decays is symmetric, its impact on observables such as $R_K$ is not symmetric because it is in the denominator and the first derivative in the error propagation is not sufficient to correctly propagate the errors. For this reason, the experimental collaborations have published profile log-likelihood values parametrized as a function of $R(K)$ to be used in phenomenological interpretations. Unfortunately, however, the Gaussian approximation is still being used in many studies. We, therefore, take the opportunity to point out that there exists open source programs that can handle these types of log-likelihood functions, e.g., Ref. [139, 140]. Below, we briefly summarize their key features.

The FlavBit module of GAMBIT [139, 140] includes custom implementations of several likelihood functions for a wide range of flavor physics observables and contains the uncertainties and correlations associated with LHCb measurements of $B$ and $D$ mesons, as well as kaons and pions. The module FlavBit uses SuperIso [141–143], which provides the theoretical predictions of the flavor observables together with their uncertainties and correlations in the SM and in BSM scenarios, and compares in detail the theoretical predictions against all relevant experimental data.

Another widely used example is the flavio [144] package. It includes a large database of experimental measurements and has an easy to use interface to carry out fits for new physics scenarios. The package allows for both frequentist and Bayesian fits and implements likelihoods via YAML files using either Gaussian approximations (with or without asymmetric uncertainties) or, where available, numerical evaluations of the profile likelihood. If more experimental information were made available in a comprehensive form, i.e., as a full likelihood, its inclusion would be straightforward and highly beneficial.

There are additional subtle points that are important to make correct interpretations of flavor anomalies. For instance, the interpretation of other measurements characterizing the $b \to s\ell\ell$ transition require theoretical input for form factors. This creates a non-trivial dependence between, e.g., fitted Wilson coefficients or new physics couplings and the theoretical input that was used.

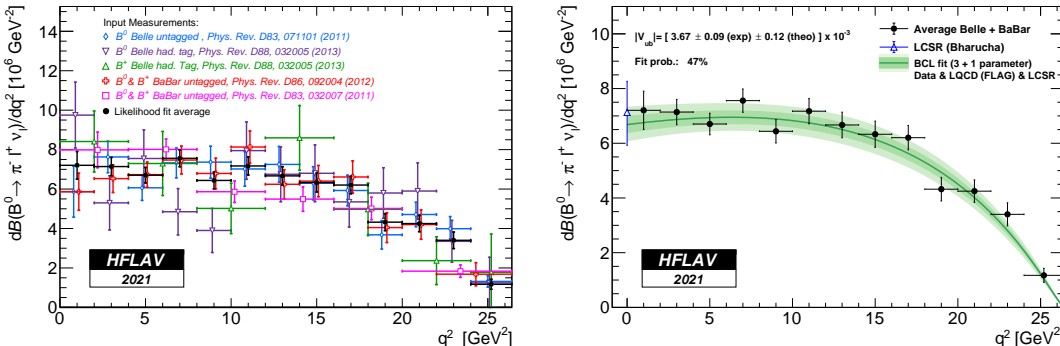

Figure 10: (Left) The measurements entering the likelihood based average from Ref. [145,146] are shown. (Right) The averaged spectrum can be used to determine $|V_{ub}|$.

### 4.4.2 Form factor and $V_{ub}$, $V_{cb}$ determinations

An excellent use case of open likelihoods are the determinations of the CKM matrix elements $V_{ub}$ and $V_{cb}$ from semileptonic $B$ hadron decays. Such measurements can be carried out with selections that yield statistically independent datasets. For instance, the most precise value of $V_{ub}$ from exclusive $B \to \pi \ell \bar{\nu}_\ell$ decays is obtained when using a global average prepared by HFLAV [145, 146] of the measured differential branching fraction as a function of the four-momentum transfer squared $q^2 = (p_B - p_\pi)^2$, shown in Fig. 10. The average is not built from published likelihoods, but from the information provided in the individual papers. Measurements are modelled using multivariate Gaussian distributions, and common systematic uncertainties are included as nuisance parameters with Gaussian constraints. If the full likelihood information, including all the relevant nuisance parameter dependencies would be available, a more consistent average could be carried out. The right panel of Fig. 10 shows the global fit for exclusive $V_{ub}$, combining experimental information with predictions about the decay dynamics from lattice QCD and QCD sum rules. The experimental data constrains the form factors at low $q^2$ values, precisely the region in which lattice QCD currently has little predictive power.

A similar work program on measurements is taking shape for exclusive $V_{cb}$ determinations. There the most precise determinations can be obtained by studying $B \to D^* \ell \bar{\nu}_\ell$ transitions. Contrary to the measurements of $B \to \pi \ell \bar{\nu}_\ell$ the focus of most measurements was to determine form factor parametrization values and $|V_{cb}|$. Most of the results use a very specific simplified parametrization, first promoted in Refs. [147, 148], and the focus of studies was largely in averaging such results. Recent studies, however, seem to indicate that with the current experimental precision the assumptions that underpin this simplification need to be re-evaluated (see, e.g., Refs. [149–152] and references therein). Unfortunately, this realization and the fact that experimental data were not preserved in likelihood form renders the results of a decade-long measurement campaign unusable for more model-independent future interpretations. This could have been avoided, if the results would have been reported with their full likelihood or presented in a manner similar to that for $B \to \pi \ell \bar{\nu}_\ell$. This is a stark example of how the potential of excellent scientific work cannot be fully realized.

### 4.4.3 Future global EFT fits for $b \to c \tau \bar{\nu}_\tau$

Another important use case are combinations of LHCb and Belle II measurements that target $b \to c \tau \bar{\nu}_\tau$. There, a persistent anomaly over the SM expectation is reported in final states involving $B \to D$ and $B \to D^*$ transitions, typically reported in ratios to $\ell = e, \mu$ as

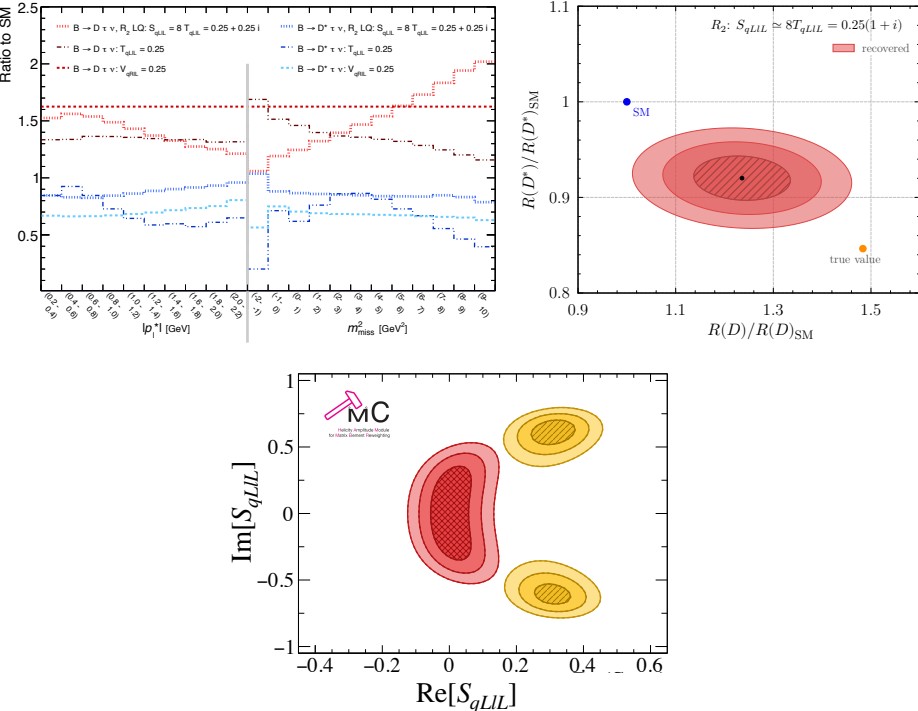

Figure 11: (Top left) The ratio of SM to new physics branching fractions for $p_\ell$ and $M^2_{\mathrm{miss}}$ are shown for scalar, vector, and tensor contributions. (Top right) The bias of simplified reinterpretations of the measured ratios $R(D^{(*)})$ is shown. All present-day measurements assume the SM nature of the underlying processes. (Bottom) The unbiased extraction of Wilson coefficients is demonstrated. This is a showcase for what can be achieved if full likelihoods and additional information would be published. All plots taken from Ref. [56].

$R_{D^{(*)}} = \mathcal{B}(B \to D^{(*)} \tau \, \bar{\nu}_\tau)/\mathcal{B}(B \to D^{(*)} \ell \, \bar{\nu}_\ell)$. All existing measurements use kinematic quantities, e.g., $q^2 = (p_B - p_{D^{(*)}})^2$, $p_\ell$, or $m^2_{\mathrm{miss}} \simeq m^2_\nu$ to discriminate signal processes from background. This encoded dependence makes it challenging to carry out new physics interpretations using the measured ratios and to probe for generic new physics explanations using an EFT approach, where contributions from ten distinct operators with complex couplings have to be considered. The dependence on the shape of the distribution of variables on scalar, vector, and tensor new physics contributions is shown in the top left panel of Fig. 11. The panel on the top right illustrates the potential bias that can arise from carrying out a simplified interpretation of just the measured ratios of $R_{D^{(*)}}$ using a toy model. The recovered value of the coupling is strongly biased.

A promising approach to constrain new physics contributions without such biases, is to directly fit the Wilson coefficients and leverage the dependence on kinematic variables (see, e.g., Ref. [153]). The constraints obtained on such Wilson coefficients can be directly interpreted within specific new physics models that provide mappings from the theory model parameters to the Wilson coefficients or to test without biases explanations that also encompass the anomalies observed in $g - 2$ or $b \to s\ell\ell$. A particularly promising approach for such global EFT fits would use the full likelihoods to combine experimental information across different channels and experiments. One possible path towards this is outlined in Ref. [56], where the experimental tool to produce reliable and fast prediction functions for arbitrary new physics is outlined. Baryonic measurements from LHCb or future measurements from Belle II prob-

ing $\tau$ properties could produce strong orthogonal constraints in the 20-dimensional coupling space. An example of such a direct Wilson coefficient fit, using a single channel, is shown in the bottom panel of Fig. 11. The injected new physics or SM coupling can be recovered in an unbiased way.

### 4.4.4 DNNLikelihood example: global EFT fits for $b \rightarrow s$ transitions

As mentioned above, global fits of $b \rightarrow s\ell^+\ell^-$ transitions exploit experimental data on $B \rightarrow K^{(*)}\ell^+\ell^-$ and $B_s \rightarrow \phi\ell^+\ell^-$ branching fractions, angular distributions and Lepton Flavor Universality Violation ratios. The theoretical calculation of these observables involves, in addition to the SM parameters, a large number of nuisance parameters, some of which, for example form factors, can be estimated using non-perturbative methods such as lattice QCD or QCD sum rules, while others, such as long-distance charm loop contributions, can be either estimated or extracted from data. As an example of global fits we consider the analysis of Ref. [154] in the framework of the SMEFT. The likelihood obtained from the Bayesian fit performed in Ref. [154] is a function of 83 parameters, of which 6 are parameters of interest, namely the SMEFT Wilson coefficients, and 77 are nuisance parameters. The likelihood is not an analytic function, as it contains several integrals, which makes the calculation of the likelihood resource-intensive. In particular, the likelihood cannot be approximated with a Gaussian (not even by adding further moments) since it contains several multi-modal directions and complicated correlations. This is an example of why a fast approximation of likelihoods would clearly be useful. Tools to produce fast approximations of statistical models and likelihoods are the kinds of developments that can be expected when the publication of full statistical models and likelihoods becomes standard practice.

One way to do speed up the calculation of the likelihood described above is to use the `DNNLikelihood` [48] package, which approximates likelihood functions using a fully-connected deep neural network (DNN). A total of $1.9 \times 10^6$ samples were obtained through an MCMC sampling of the likelihood, which was used to define training ($10^6$), validation ($5 \times 10^5$) and test ($4 \times 10^5$) sets for the `DNNLikelihood`. We note that the `DNNLikelihood` package (in the final stage of its development) was implemented and optimized in `Python`, required limited effort and did not need extreme tuning of hyperparameters to reach the required performances. Once a good DNN model was found, the final training required a few hours on a Tesla V100 GPU, the resulting `ONNX` model file has a size of about 10 megabyte, and the point-by-point calculation time of the DNN approximation to the likelihood is around $2.6 \times 10^{-6}$ seconds, that is, $\mathcal{O}(100)$ to $O(1000)$ times faster than the original likelihood. In Fig. 12 we show the 2D marginal probability distribution of the parameters of interest and some nuisance parameters obtained with the original test samples (green) and with the `DNNLikelihood` (red). Agreement up to the 3-standard deviation level is excellent, showing that the `DNNLikelihood` offers a fast, flexible, and accurate, approximation to the original likelihood. Moreover, this study illustrates once again the importance of detailed likelihood information for reliable fits.

## 4.5 Dark matter direct detection

Direct dark matter experiments search for dark matter particles interacting within a detector, employing a number of target materials and technologies. The typical challenges driving the statistical treatment of these experiments include sometimes incomplete knowledge of backgrounds, and, typically, very low event rates necessitating exact methods or simulations to perform inferences. For WIMP dark matter searches, in particular, for higher-mass ($> 6 \, \text{GeV}/c^2$) WIMP searches using liquid noble-gas detectors, recommended procedures for performing inference, primarily with the profile likelihood ratio, and choosing common (constant) astrophysical parameters have been published [155].

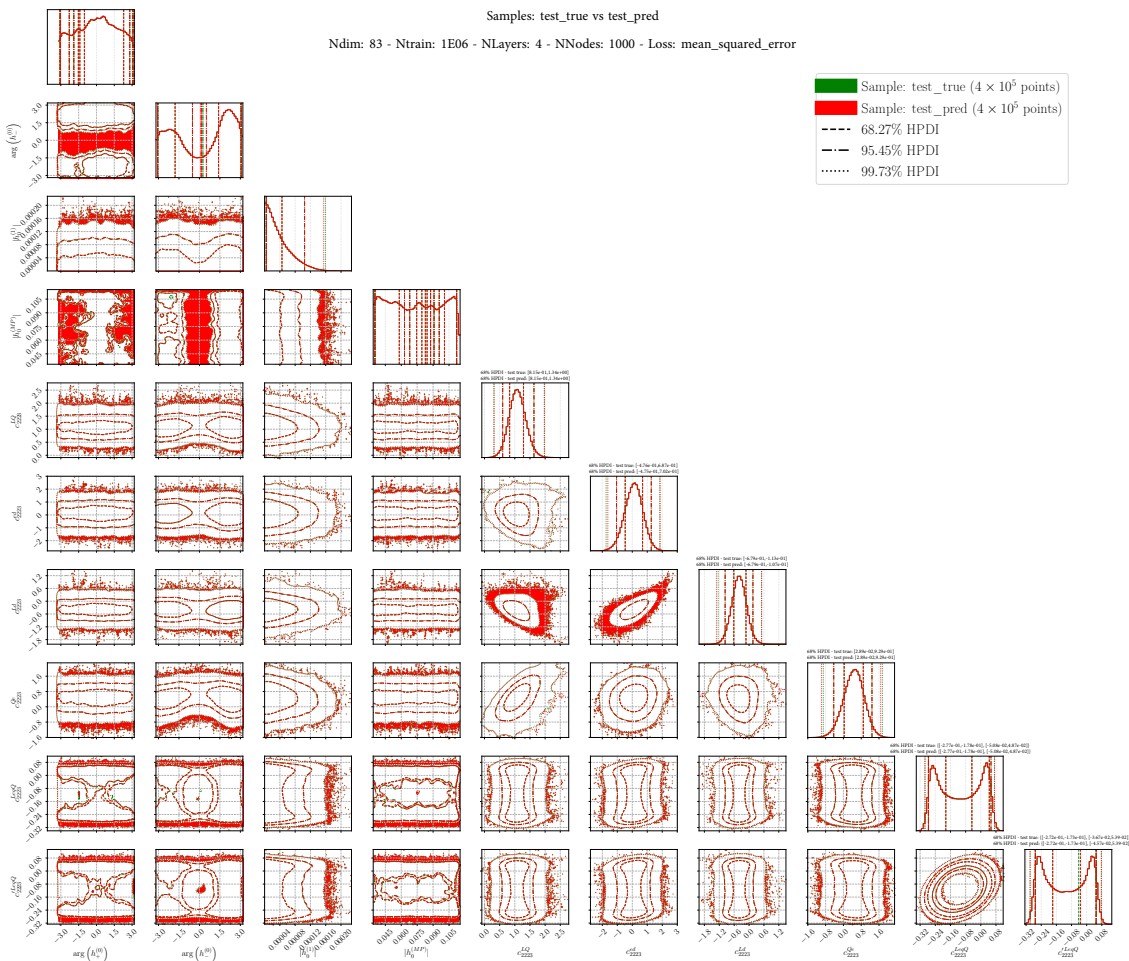

Figure 12: 1D and 2D marginal probability distributions for the six parameters of interest and for four selected nuisance parameters from the `DNNLikelihood` global fit of $b \to s\ell^+\ell^-$ transitions from Ref. [154]. For the parameters of interest the 68% Highest Probability Density Interval (HPDI) is reported on top of the 1D distributions.

In addition to detector-specific parameters, results will depend both on astrophysical parameters such as the assumed dark matter density or velocity distribution, nuclear physics form factors and yield parameters common for certain target materials. Unless the scaling is trivial, such as changing the local dark matter density, reinterpreting experimental results will require the full statistical model used by the experiments. In the following sections, we describe two such efforts, one by the CRESST-III Collaboration, and the other by XENON1T, specifically their ionization-only analysis.

### 4.5.1 CRESST-III public results

In the CRESST experiment particle interactions are measured by means of scintillating calorimeters. The conventional target material is $CaWO_4$ in form of crystals. An incoming dark matter particle may collide with any of the individual nuclei (with probability proportional to its respective interaction cross section). The data consist of a set of 2D points representing the measured deposited energy in the detector and the scintillation efficiency, Light Yield, associated with the particle interaction. The Light Yield permits the identification of the nuclear recoils and it is used to define the region of interest (ROI). For a given model $f(E_R|\omega)$, the

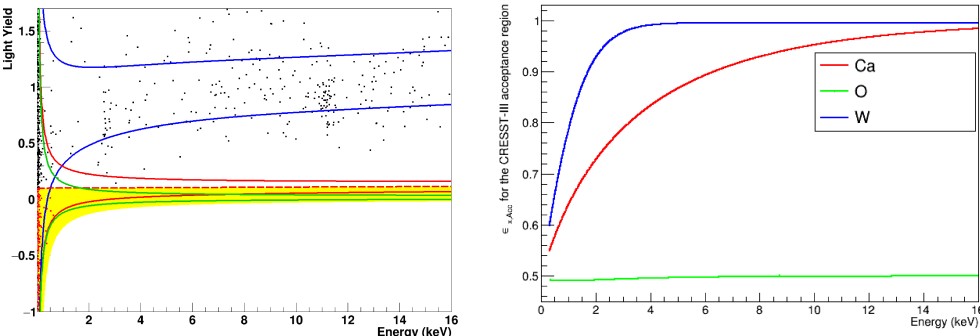

Figure 13: Analysis space and efficiencies for the CRESST-III dark matter search. Left: Electron (blue), oxygen (red), and tungsten-recoil bands (green). The bands have been fitted to the neutron calibration data and are required for the definition of the acceptance region (yellow) in the blind data. Right: The fraction of accepted events in the nuclear recoil bands, per nuclear species as a function of the reconstructed energy. Plots taken from [156].

probability for a nuclear recoil of energy $E_R$ to be reconstructed at energy $E_{\text{reco}}$ reads

$$
\begin{aligned}
f(E_{\text{reco}}|\omega) = \Theta(E_{\text{reco}} - E_{\text{thr,reco}}) \cdot \tilde{\varepsilon}(E_{\text{reco}}) \cdot \varepsilon_{\text{acc}}(E_{\text{reco}}) \\
\times \int_0^{+\infty} f(E_R|\omega) \mathcal{N}(E_{\text{reco}} - E, \sigma^2) dE \,,
\end{aligned}
\tag{15}
$$

where $\Theta(E_{\text{reco}} - E_{\text{thr,reco}})$ is a hard cut representing the trigger (only energies above $E_{\text{thr,reco}}$ are taken into account), $\tilde{\varepsilon}(E_{\text{reco}})$ is the fraction of events surviving the cut selection, $\varepsilon_{\text{acc}}(E_{\text{reco}})$ the fraction of events occurring in the acceptance region in the Light Yield - Energy plane and $\mathcal{N}(E_{\text{reco}} - E, \sigma^2)$ is a normal distribution accounting for the finite energy resolution $\sigma$ of the detector. In Ref. [156] the CRESST Collaboration published the measured recoil energies of the result in Ref. [157]. Together with the recoil data, selection and acceptance efficiencies are made available, as shown in Fig. 13.

The likelihood, a special case of Eq. (7), can be written as

$$
L(\omega \,|\, \{E_{\text{reco},i}\}) = \frac{e^{-N_{\text{exp}}} N_{\text{exp}}^{N_{\text{obs}}}}{N_{\text{obs}}!} \prod_{N_{\text{obs}}} f(E_{\text{reco},i}|\omega) \,,
\tag{16}
$$

where $N_{\text{exp}}$ is the mean (expected) number of nuclear recoils under the model assumptions, and $N_{\text{obs}}$ the number of observed events.

For the results in Ref. [157] no assumption about the background has been made and the sensitivity to the spin-independent, coherent, elastic scattering off nuclei by dark matter particles was determined by means of the Yellin optimum interval method [158]. All events in the acceptance region have been (conservatively) taken to be potential signal (that is, $f(E_R|\omega)$ does not contain a background model). The information encoded in Eq. (16) allows for model testing and other inferences and may be updated by adding a background model.

### 4.5.2 XENON1T S2-only data

The XENON1T Collaboration has released the statistical model used in a charge-only analysis [159]. The energy threshold for producing a charge signal in a liquid xenon time projection chamber is significantly lower than the standard charge+scintillation light signature, down to

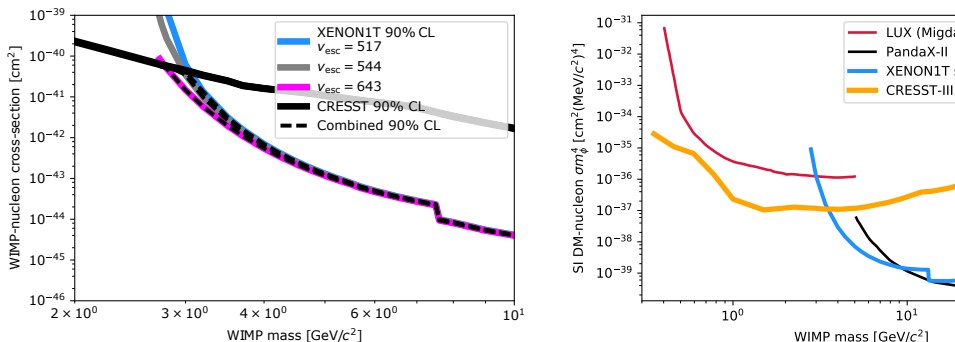

Figure 14: (Left) Constraints on the spin-independent WIMP-nucleon cross section using CRESST-III [157] and XENON1T ionization-only [159] data, altering the assumption on the Milky Way escape velocity $v_{\text{esc}}$. Results from CRESST-III (black), with a lower energy threshold are approximately constant, while the higher energy threshold means the XENON1T limit is weakened with a lower escape velocity. The dashed black line shows a combined result. (Right) Spin-independent WIMP-nucleon cross sections in the case of a light mediator mass. The CRESST-III result has be determined according to Ref. [159] making use of the data public release, LUX results are presented in Ref. [161], PandaX-II in Ref. [162].

0.2 keV electronic recoil energy. Many new physics models could be probed in this range, including dark matter-nucleus scattering, dark matter-electron scattering, axion-like particles and dark photons. The background model of this analysis is incomplete, and the signal response has large uncertainties. The statistical treatment of the data was performed counting events in a region of interest, with 30% of the observed data used as a training set to optimize the region for each signal hypothesis, and the remaining 70% to set the limit. The expected number of events with a certain ionization energy,

$$ f(S2) = \int_0^\infty \epsilon(S2|E_{\text{recoil}}) f(E_{\text{recoil}}) \, \mathrm{d}E_{\text{recoil}} \, , \tag{17} $$

is given by integrating the recoil spectrum $f(E_{\text{recoil}})$ weighted by the response matrix $\epsilon(S2|E_{\text{recoil}})$, where $E_{\text{recoil}}$ is the true recoil energy and $S2$ is the observed ionization signal. In practice, $\epsilon$ is described by a finely binned 2-dimensional histogram.

Public results are available, together with software to compute upper limits [160], and include: the response matrix, published results in the optimized regions used by the collaboration, (known) background estimates and the training and science data. Together, this allows both fast recasts of results using the already-optimized search regions for spectra close to one of the published results, and more elaborate recasts including using the training data to define new optimal intervals for any spectrum.

### 4.5.3 Example combination

Open statistical models in direct detection will allow combining results from multiple experiments either with different and complementary detector technologies or with shared nuisance parameters for similar experiments. Assumptions on the dark matter velocity distribution can be updated as new and more precise data becomes available, and signal models can be easily extended beyond the customary spin-dependent and spin-independent WIMP-nucleon interactions. As an example, a low-threshold detector with a higher background rate may publish a higher upper limit than a higher-threshold detector with a low background but be more robust

to uncertainties in the modelling of the dark matter velocity distribution tail. A simple example is shown in Fig. 14 (Left), where the XENON1T result depends on the galactic escape velocity. In this case, the different detector responses mean that for most of the parameter space, one or the other detector dominates the limit, but if every collaboration provided the necessary statistical model, regions where several detectors are competitive could be better constrained by a combined limit, thereby strengthening the final results. In Fig. 14 (Right) predictions of light mediator models for different experiments are reported together. The CRESST-III sensitivity (not present in the data release) was determined using the data release and interpreted in the light of the light mediator models developed for the XENON1T analysis. Having the full likelihood available allows for the determination of the sensitivity at any desired confidence level. In the example presented in Fig. 14 (Left), the sensitivities (at each WIMP mass) of the individual searches were chosen such that the combined $p$-value equals 0.1.

## 4.6 World averages

A special case of the reinterpretation of data are averages of measurements. These are often done by large averaging collaborations, such as the Particle Data Group (PDG) [76, 163] or the Heavy Flavor Averaging Group (HFLAV) [145, 146] which succeeded the LEP Heavy Flavor Steering Group. The combination of measurements from the statistical point of view is similar to that of the global BSM fits to be discussed in the next section. The difference is that instead of a BSM model we are combining multiple measurements of one or several observables without any theory interpretation.

As discussed in the previous sections, one of the main challenges is absence of sufficient publicly available information. The problem faced by the HFLAV group in the determination of a world average of $V_{cb}$, involving theory input, was mentioned in Section 4.4.2. But the problem of insufficient information arises already in the simple case of averaging multiple measurements of a single parameter, for example, a branching fraction. The likelihood function of a measurement is reasonably well defined if a central value (a point estimate) with symmetric uncertainties is quoted. The situation is already problematic in case of a result with asymmetric uncertainties, as discussed for $R(K)$ in Section 4.4. Such a result defines the log likelihood at three points only: at the point estimate and at the lower and upper bounds defined by the asymmetric uncertainties. For any other parameter values assumptions have to be made about the shape of the likelihood function.

If a limit on the parameter is published the situation is even worse. The likelihood function of the measurement is largely undefined. The only available information (in case of a Bayesian credible interval) is that the integrated likelihood over a quoted range has a certain value.

A further, common, example of insufficient information is when (external) input parameters were used in a measurement but their values are not quoted in a publication. Sometimes certain assumptions were made in some measurements, but not in others or assumptions differ in different measurements. Such information is essential in order to perform a valid combination of measurements.

Another area of frequent difficulties are correlations. Correlations between measurements by different collaborations as well as correlations between measurements by the same collaboration with different methods or datasets must be known if correct averages are to be produced. Unfortunately, this information is often not or only partly available. It is for this reason that the averaging collaborations are composed of members of various experimental collaborations who may be able to obtain the required information.

Correlations arise from common parameters. While these are usually well defined for physics parameters, such as daughter particle branching fractions, the situation is often more challenging for nuisance parameters describing systematic effects. For example, consider the trigger or reconstruction efficiencies for two datasets taken with evolving detector perfor-

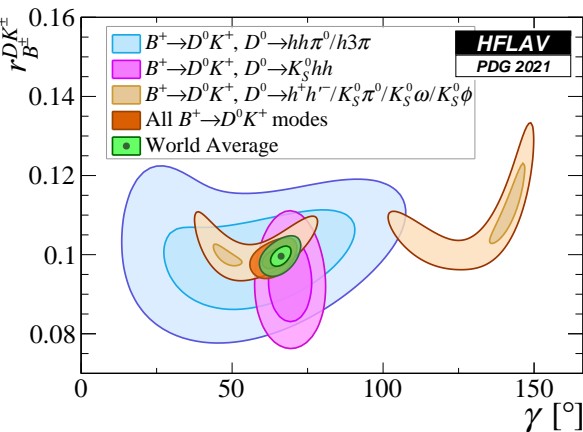

Figure 15: Constraints on the CKM unitarity angles $\gamma$ and the ratio of decay amplitudes $r_{B^\pm}^{DK^\pm}$ obtained from the combination of multiple measurements [145,146] with the `GammaCombo` framework [164,165].

mances or configurations or processed with different reconstruction algorithms. Quantifying the correlation can be hard even for those with access to internal information.

The systematic publication of likelihoods would address major problems in the combination of measurements. It has been noted earlier in this paper, and stressed again here, that it is not sufficient to have the full likelihood as function of all parameters; it is also necessary to publish meaningful explanation of the parameters, in particular, the nuisance parameters, so that those which induce dependencies between measurements can be readily identified. Only if there is a common understanding of the meaning of the parameters can they be treated correctly in the combination of multiple measurements. Developing procedures and tools to achieve this is a major effort that will get more attention with the increasing availability of open likelihoods.

A further aspect to point out is that the averaging collaborations are not only consumers, but also producers of likelihoods. In particular, in case of non-trivial correlations between averages a likelihood function provides the best way of publishing the results. As an example, Fig. 15 shows the current way of publishing an averaging result in the form of a plot. The non-Gaussian shape of some averages is evident. Anyone who wishes to use such an average would have to extract the contour lines from the plot and extrapolate the likelihood for parameter values that are not on the contour lines. Clearly, the community at large would benefit if, first, the World Averages, as is true of global fits in general, can be done more reliably based on the full statistical models; second, their results are communicated in the form of multi-dimensional likelihoods, which can easily be reused, and, third, a common understanding of parameters can be achieved.

## 4.7 BSM global fits

BSM global fits are statistical analyses of the entire parameter space of new physics models taking into account all relevant experimental data. The goals are to understand which models are allowed or favored by data, which regions of their parameter spaces are allowed or favored, and what the models predict for future experiments [166]. BSM global fitting began in earnest just before the first run of the LHC [167–179], motivated by the need to understand the discovery prospects for models of new physics and to optimize search strategies for new particles. The multi-dimensional fits were possible at that stage as software to compute predictions from popular models throughout their parameter spaces reached maturity, and the field

caught up with advances in computing power and the MCMC computational revolution of the 1990s [180], as well as other techniques for the exploration of many-dimensional parameter spaces [179].

In the last few years, in particular, global fits turned into a challenging undertaking, due to the breadth of available data from particle physics, astrophysics and cosmology, their associated uncertainties, and the breadth of new physics models under consideration. State-of-the-art BSM global fits are now performed by large collaborations comprising experimental and theoretical physicists, using dedicated software frameworks such as GAMBIT [139], MasterCode [181], HEPFit [182] and Fittino [183].

As the preceding sections and, in particular, the examples in Figs. 5, 6, 8 and 11 show, insufficient likelihood information can significantly distort the result of a parameter estimation analysis. Since a BSM global fit aims at a statistical combination of all experimental results relevant for a new theory — from measurements of the Higgs boson, heavy flavor and electroweak precision observables, to searches for new particles at the LHC and in dark matter experiments — these likelihood challenges compound in global fit analyses. In short, constructing all the necessary likelihoods from limited publicly available information is both time-consuming and error prone, and inevitably involves approximations that reduce the impact of the data. Indeed, it was noted as early as 2006 [173] that global fits would benefit from open likelihoods.[16]

Whilst some choices in the construction of likelihoods from public information seem innocuous, they may not accurately approximate the true likelihood, e.g., using a Gaussian error model for experimental measurements summarized by a point estimate (a central value) and an uncertainty, such as the combined Higgs boson mass measurement reported by the Particle Data Group [76]. This choice may be reasonable under certain regularity conditions due to asymptotic results (see, e.g., Ref. [16]) or justified as the least informative choice compatible with the available information (see, e.g., Ref. [184]). As discussed in Section 4.6, a similar situation arises when an experimental result is communicated only as an upper or lower limit, from which an approximate likelihood function must be constructed (see, e.g., Ref. [139] for a discussion of some of the different choices in constructing such likelihoods).

The absence of open statistical models not only reduces the impact of individual experimental results in constraining a BSM theory, it also limits the possibility to fully exploit the complementarity between different results. A clear manifestation of this is the case of likelihoods for LHC searches for new particles. Here we point out why this presents a particular challenge in the context of global fits.

As also discussed in Section 4.3, most LHC analyses define multiple SRs, and for many analyses the published information is insufficient to reconstruct a statistical model that accounts for the (background) correlations between them. In such cases, the standard approach is to consider only the most sensitive signal region for each BSM parameter point. Discarding the information from all the other SRs is clearly undesirable, and particularly so in the context of BSM global fits. These fits typically study fully-developed BSM theories with many free parameters and rich phenomenologies, such that a single parameter point might predict signal contributions across multiple, and very different, signal regions. Without sufficient public likelihood information (or, better, the full statistical models, see Section 4.3) these wide-ranging signal predictions cannot be confronted with the full constraining power of the data.

Finally, insufficient likelihood information induces a numerical stability problem that currently restricts the scope of global fits with full MC simulation of LHC analyses, such as in Ref. [137]. When only the most sensitive SR from each LHC analysis can be used, there will be jump discontinuities in the likelihood function between regions of parameter space in which

---

[16]We note that the same fundamental challenge of limited open likelihood information also affects other classes of global fit analyses, such as fits of the three-flavor neutrino sector.

different SRs are chosen. These occur even when SR sensitivities are equal at the parameter region boundaries, as the likelihoods in the respective SRs depend on the observed and expected event counts. With signal predictions from MC simulations, the SR sensitivity estimates are noisy. This then leads to particularly noisy estimates of the likelihood, as the SR choice fluctuates between SRs of similar sensitivity but quite different likelihood. In a global fit that optimizes the likelihood, fluctuations that increase the likelihood are retained, biasing the statistical inferences. To tame this problem typically requires far larger event samples per point than would be necessary if all SRs could be used simultaneously, and the associated computational cost severely restricts the size of BSM parameter spaces that can be explored in such fits.

In the above case, access to more complete likelihood information can remove a severe computational hurdle. But, in general, the many nuisance parameters, which full likelihoods and statistical models inevitably introduce, will increase the computational challenge of BSM global fits. Therefore, it is important to explore strategies for tackling this, including the use of pruned and/or partially profiled/marginalized statistical models, GPU parallelization and fast surrogate models such as that described in Section 4.4.4 to speed up computations, and *fast-slow* parameter sampling algorithms.[17] In all cases, having access to the full statistical models will be of key importance to determine the appropriate balance between approximation and computational cost for a given global fit.

While Bayesian and frequentist methods are common in global fits, the latter are almost exclusively likelihood based, with frequentist results utilizing asymptotic formulae (e.g., Wilks' theorem [188, 189]) in place of the sampling distribution. As the full likelihoods are often intractable, however, there is scope for simulation-based inference using the full statistical model in both frequentist and Bayesian settings. Furthermore, similar to the *p*-value computations discussed in Section 4.3 for combining LHC search results, having a full statistical model for the data will also greatly aid efforts to compute global *p*-values in BSM global fits since the distribution of the fit test statistic can then be reliably determined by simulation; see Refs. [190, 191] for efforts in this direction.

Before closing this section, we note that in addition to *using* public data products including statistical models, global fits *create* them. The results of a global fit, be it a collection of posterior samples created using MCMC sampling or otherwise, are valuable and computationally expensive to obtain, and could be preserved for future scrutiny and reuse. See, e.g., Refs. [192, 193] for examples published by the GAMBIT Collaboration using Zenodo. In summary, open statistical models from particle physics, astrophysics and cosmology would make it easier and faster to perform reliable global fits, alleviate numerical pathologies introduced by otherwise necessary approximations, enable simulation-based approaches, and, lastly and perhaps most importantly, allow fits to harness the full power of the breadth of experimental data available to us today and in the decades to come.

# 5 Challenges and outstanding issues

We recognize that the publication of statistical models represents a significant change in the status quo for the particle physics community. Compared to highly processed results in tables and figures, the statistical models expose more of the complexity of the analyses. There is a longstanding tradition of collaborations communicating the essential ingredients of an analysis and the impact of various sources of systematic uncertainties. However, the traditional vehicles

---

[17]In fast-slow algorithms (nuisance) parameters that only affect fast computations can be varied much more frequently than the other parameters. See Ref. [185] for an application of this approach in a BSM global fit, using the `PolyChord` [186] sampler via `ScannerBit` [187].

for this information have often been more suggestive than precise. The fidelity and level of detail represented in the full statistical model exposes far more than traditional approaches, and historically the experimental collaborations have only shared this information with other experimental collaborations for narrow purposes in highly-controlled environments such as the ATLAS-CMS Higgs boson combination. However, this highly-restricted mode of sharing does not allow for what is clearly valuable science to be performed by qualified physicists who work outside the collaborations. This forces the use of unnecessary, poorly controlled, and time consuming approximations. The move to a more open model for publishing these high-value scientific artifacts will remove these barriers and lead to more, and higher-quality, science. However, we also recognize and expect that it will take time and effort to establish new norms and conventions.

There are a number of developments that will enhance, facilitate, and streamline the use of the published statistical models. Here we list a few points that we encourage the community to address in dedicated workshops in the near future.

- Unambiguous definitions and systematic naming conventions for parameters (including nuisance parameters) are difficult to achieve across analyses even within a given experiment. Nevertheless, it would be highly desirable to achieve some coherence in naming conventions, as this would greatly facilitate, for example, one-to-one comparisons and *combinations*.

- As noted in Section 2.2 and illustrated in Section 4.3, when publishing likelihood functions some care is needed to choose broadly useful parametrizations. In particular, a likelihood parametrization in terms of quantities such as masses, cross sections, widths, branching fractions, etc., is often more useful than a parametrization in terms of theory-model (Lagrangian) parameters. While likelihood functions are in principle reparametrization-invariant, meaning any convenient choice is fine as long as it is technically possible to reparametrize the likelihood later, one has to beware of introducing dependencies which can result in a loss of information.

- Questions of runtime for the evaluation of full statistical models or likelihoods can be of concern. Often, millions or billions of model parameter points need to be evaluated in, e.g., sampling-based inference. Therefore, it would be desirable to be able to evaluate likelihoods in $\mathcal{O}(\mathrm{ms})$. Currently, many large statistical models are significantly more computationally expensive. Therefore, strategies and public tools for pruning, simplification and/or partial profiling of the full likelihood model need to be developed. Efforts in this direction are underway in collaboration between the experimental and the theory communities.

- Currently, PDF fits (and similar uses of the data) are largely based on procedures that assume Gaussian error sources. Making it possible to capitalize on the availability of full likelihoods will require considerable effort to implement a more sophisticated fitting procedure based on the minimization of a negative log-likelihood or efficient sampling of the likelihood parameter space. It can be expected, at least initially, that some simplifications will be necessary, or at least very helpful. We also note that at present many fits (PDF, EFT, global, etc.) deal with very large correlation matrices with a high degree of redundancy in the nuisance parameters, so that some pruning would be appropriate.

- Although this paper has been focused on communicating the statistical models used in HEP analyses rather than on their underlying structure and assumptions, we mention here a possible refinement that could be considered by statistical modelers. The Gaussian models widely used in HEP analyses usually assume that the standard deviations are

exactly known. In practice, this is often an inappropriate assumption, particularly for systematic errors, which are often obtained through approximate procedures and may even reflect educated guesses. It is possible in such situations to treat the true, unknown, variances as adjustable parameters, and to model the estimated variances as random variables. For example, in Ref. [194] the latter are modeled using a gamma pdf. In this type of model, the size of confidence intervals becomes dependent on the internal consistency of the data and the sensitivity of fits and averages to outliers is reduced. A Bayesian approach to the same problem is discussed in Refs. [195, 196].

Open science is gaining momentum worldwide and it is fundamentally tied to the health of our field. CERN and other labs are embracing open science and crafting policies accordingly, and the establishment of the FAIR principles is an important part of this development. One of the most important aspects of the idea behind these principles is that publishing knowledge in a usable and open format is not only a means for solving an immediate need, but a goal in itself. From this knowledge, new science will grow in foreseeable and unforeseeable ways.

Particle physicists have the means to start publishing full statistical models of measurements now, even if the full potential of their use might not be fully realized until new and better tools and conventions have been developed. Asking for these tools and conventions to be fully developed before starting to routinely publish full statistical models would hinder their open and rapid development. It would effectively mean that information that could be saved now would be lost as analysis teams move to new tasks or experiments.

# 6 Summary

The statistical model underlying an experimental result is essential information as it describes the probabilistic dependence of the observable data on the parameters of interest and the nuisance parameters. Therefore, we advocate that publication of the full statistical model together with the corresponding observed data become standard practice. Technical solutions now exist to make this feasible.

The published statistical model should include as distinct, identifiable, components the terms pertaining to the primary measurements as well as the auxiliary measurements as described in Section 2.2. This allows one to evaluate the full likelihood and reproduce common results, including measurements and constraints on models of new physics. It also enables combinations within and across experiments. Moreover, access to the full statistical model enables the generation of pseudo data (toy MC) needed for frequentist $p$-values and confidence intervals. While more limited, we also recognize the utility of publishing (profile) likelihoods in a pertinent parametrization and recommend that this practice continue.

Ideally, the statistical models and likelihoods would be published in a serialized (written to a file) format that is both human-readable and machine-readable with a declarative specification that provides a mathematical definition of the model. We elaborated on the technical aspects pertaining to this; in particular, we discussed tools and general considerations around serialization and down-stream use in Section 3.

The available details of a statistical model may heavily affect the short- and long-term impacts of any measurement. We discussed a number of physics use cases from across the field, ranging from PDF to EFT fits, from constraints on new physics to the determination of Wilson coefficients, and from world averages to BSM global fits, which show how the publication of more precise likelihood information and, ideally, the full statistical models could significantly enhance the impact of experimental results. These use cases, which are by no means exhaustive, constitute a clear call to action.

An immediate action that can be taken by the community: (*i*) publish all the associated `RooWorkspaces` or (*ii*), for binned statistical models based on the `HistFactory` specification, publish the models in the `pyhf` JSON format. This would provide the impetus for the development of tools to make the use of the published models user-friendly, efficient, and effective. Longer-term developments are certainly needed to enhance, streamline, and facilitate the use of published statistical models. However, the publication of the currently available statistical models would already be a watershed development in the field, one we hope the community is ready to embrace.

# Acknowledgements

This work was supported in part by the National Science Foundation under Cooperative Agreement OAC-1836650 (KC, MF and MN); the U.S. Department of Energy Award No. DE-SC0010102 (HP); the IN2P3 master project "Théorie – BSMGA" (SK); the Buchman Scholarship Funds and the Azrieli Foundation (IB); the Knut and Alice Wallenberg Foundation and the Swedish Research Council (JC); the Science and Technology Facilities Council, STFC, through grants ST/S000844/1 (GC), ST/T000856/1 (RT) and ST/N003985/1 (NW); the DFG - Germany's excellence strategy EXC - 2094-390783311 (NFI); the NSFC Research Fund for International Young Scientists grant 11950410509 (AF); and the National Science Foundation under Award no. 1719286 (KDM).

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
