# Peer review of "Publishing statistical models: Getting the most out of particle physics experiments"

_SciPost Physics, doi:SciPost Phys. 12, 037 (2022)_

## Round 1 · Referee Report · Anonymous · 2021-10-17

Strengths

1. Clear and concise description of the statistical language
2. Comprehensive and clear list of examples that demonstrate clearly the potential consequences of information loss on repeating, re-evaluation, combining, recasting or reinterpreting results.

Weaknesses

1. over-use of jargon that may limit the accessibility of the message outside the immediate collider physics community
2. Confusing structure to Section 3 and perhaps a little too much detail

Report

I enjoyed reading this paper and found section 4 particularly interesting and enlightening. Section 2 gave an excellent and concise description of the statistical language used that is particularly important when terms are often used in colloquial fashion.

The overall message of the paper is a strong one and the examples in Section 4 show that there are real advantages to publishing/providing the full statistical model when releasing results.

I did find Section 3 to be somewhat less clear and more difficult to read than the rest of the paper. I would have expected slightly more content in the introductory paragraph, briefly outlining the key technical considerations and the issues relating to open-world and closed-world models - then followed by the summary of the approaches/tools being used currently to address these considerations.

Instead there is a great deal of (perhaps overly detailed) information about RooStats and RooFit and the focus seems to be on a description of the features of those technologies rather than the technical considerations that shaped their design and development. The result of this is that the technical considerations end up being somewhat hidden in the exposition.

With reference to the publication acceptance criteria I found that the paper meets the general acceptance criteria (subject to a few minor points listed in the requested changes below).

However, I struggled to see how it relates to the "expectations" for SciPost Physics. I couldn't see a strong alignment with any of the key expectations.

"Expectations (at least one required) - the paper must:

Detail a groundbreaking theoretical/experimental/computational discovery;

Present a breakthrough on a previously-identified and long-standing research stumbling block;

Open a new pathway in an existing or a new research direction, with clear potential for multipronged follow-up work;

Provide a novel and synergetic link between different research areas."

The message of the paper is important and compelling but strictly speaking does not appear to be well targeted at this journal. The strongest overlap appears to be to the third element:

"Open a new pathway in an existing or a new research direction, with clear potential for multipronged follow-up work;",

Perhaps one could argue that although the idea of publishing full statistical models is far from new - and far older than the Les Houches recommendations from 2012 - and although a specific pathway to do this has existed for quite some time - that perhaps the fact that few in particle physics seem to have taken the pathway means it is high time to point out the pathway again and its advantages with some key examples.

On balance - the paper should be published and might be appropriate for SciPost Physics given the importance of the message. But I would welcome comment from the Authors on how they see the alignment with the editorial guidance.

Requested changes

1. Page 6: the first type of reparametrizing - suggest drop the reference to efficiencies and acceptances - the key point is that the distributions don't change. Efficiencies and acceptances could change without changing distributions and other things could change that would change the distributions. I think it is sufficient to state that the distributions do not change.

2. Page 7: "In some analyses the parameters of interest represent the expected numbers of entries in bins of a differential distribution. There are two basic approaches to this problem..." The second sentence refers to a problem - but the first sentence which is the definition of "the problem" isn't clear about what the actual problem is. This could be made clearer.

3. Figure 3 - has insufficient explanation about the different contributions to plot - it would be good to add some text to explain that these represent different contributions from different samples to each of the signal regions - and it also contains the mysterious "SR_metsigST" which could probably be removed.

4. Page 21: "This is serious limitation" ->"This is a serious limitation."

5. Figure 7 - The right panel clearly includes a plot with the words "ATLAS Internal" in violation of ATLAS rules. A quick visual inspection of the corresponding plot in the reference suggests this is the same plot - but the official public version should be used.

6. Page 28 - please add a brief description of SModelS

7. Some of the links in the bibliography seem broken - specifically those that are in the form of the arxiv number "2010.00356". This might be a client issue on my side - but please double check.

8. Section 3 could use either some restructuring or at least some additional text in the introductory paragraph that outlines what the specific technical considerations are as well as a concise definition of the terms "open world" and "closed world".

  • validity: high
  • significance: high
  • originality: good
  • clarity: high
  • formatting: excellent
  • grammar: perfect

Author:  Sabine Kraml  on 2021-11-19  [id 1956]

(in reply to Report 1 on 2021-10-17)

We thank Referee 1 for the overall very positive assessment of our paper. Regarding the referee’s remark about alignment with the key expectations of SciPost Physics, we think that from the list given at https://scipost.org/SciPostPhys/about#criteria

Expectations (at least one required) - the paper must: 1- Detail a groundbreaking theoretical/experimental/computational discovery; 2- Present a breakthrough on a previously-identified and long-standing research stumbling block; 3- Open a new pathway in an existing or a new research direction, with clear potential for multipronged follow-up work; 4- Provide a novel and synergetic link between different research areas.

we meet in fact Criteria 2, 3, and 4:

Criterion 4: Section 4 of the paper is devoted to detailing how publication of statistical models positively impacts all the areas listed in it. This provides novel synergetic links between theoretical and experimental physics in these areas. Moreover, it provides novel synergetic links between these research areas, for example enabling statistically sound combinations of experimental information in global analyses (c.f. Section 4.3.3).

Criterion 3: These synergetic links indeed open new pathways for multipronged follow-up work, as also detailed in section 4.

Criterion 2: We are convinced that this is achieved here as we clearly (we hope) presented the need and desire for public probability models in the broader community and that with the publication of the first HistFactory probability models we've gone from none for 20 years to 18 in under 2 years (with more coming from ATLAS with future LHC Run 2 publications).

We also note that our paper is following up on the Reinterpretation Forum report, which was also published in SciPost Physics (SciPostPhys.9.2.022, Ref. 6 in our paper) and which noted the need for detailed likelihood information without going into the technical details that we now discuss in our paper.

Regarding the requested changes, points 1-7 are easy to accommodate, and we will follow the recommendations from the referee. We do not quite agree with the criticism on section 3, but will try to improve the introductory paragraph as suggested by the referee. A concise definition of the terms "open world" and "closed world" is already given in the “Glossary of terms” in Fig. 1.

---

## Editorial Decision

published